



# OMEN-SED(-RCM) (v1.1): A pseudo reactive continuum representation of organic matter degradation dynamics for OMEN-SED

Philip Pika[1], Dominik Hülse[2], and Sandra Arndt[1]

[1]BGeosys, Department Geoscience, Environment & Society (DGES), Université Libre de Bruxelles, Brussels, Belgium
[2]Department of Earth Sciences, University of California, Riverside, CA 92521, USA

*Correspondence to:* Philip Pika (philip.pika@ulb.ac.be)

**Abstract.** OMEN-SED is a one-dimensional, analytical reaction-transport model for early diagenesis in marine sediments. It explicitly resolves organic matter (OM) degradation and associated biogeochemical terminal electron acceptor, reduced species and nutrient dynamics in porous media under steady state conditions. OMEN-SED has been specifically designed for the coupling to global Earth System Models and the analytical solution of the coupled set of mass conservation equations ensures the computational efficiency required for such a coupling. To find an analytic solution, OMEN-SED expresses all explicitly resolved biogeochemical processes as a function of OM degradation. The original version of OMEN-SED contains a relatively simple description of OM degradation based on two reactive OM classes, a so-called 2-G model. However, such a simplified approach does not fully account for the widely observed continuous decrease in organic matter reactivity with burial depth/time. The reactive continuum model that accounts for the continuous distribution of organic compounds over the reactive spectrum represents an alternative and more realistic description, but cannot be easily incorporated within the general OMEN-SED framework. Here, we extend the diagenetic framework of OMEN-SED with a multi-G approximation of the reactive continuum model (RCM) of organic matter degradation by using a finite, but large number of OM fractions, each characterized by a distinct reactivity. The RCM and its multi-G approximation are fully constrained by only two free parameters, $a$ and $\nu$, that control the initial distribution of OM compounds over the reactivity spectrum. The new model is not only able to reproduce observed pore water profiles, sediment-water interface fluxes and redox zonation across a wide range of depositional environments, but also provides a more realistic description of anaerobic degradation pathways. The added functionality extends the applicability of OMEN-SED to a broader range of environments and timescales, while requiring fewer parameters to simulate a wider spectrum of OM reactivities.





# 1 Introduction

The degradation of organic matter (OM) in marine sediments is a key component of the global carbon cycle and climate (Berner, 1991; Archer and Maier-Reimer, 1994; Ridgwell and Zeebe, 2005). It is the main engine behind the complex and dynamic network of biogeochemical reactions in marine sediments and, thus, controls benthic carbon and nutrient recycling,

as well as carbon burial (e.g. Arndt et al., 2013; LaRowe et al., 2020b). As a consequence, it exerts an important influence on marine primary production, the strength of the primary long-term sink for atmospheric $CO_2$ and, ultimately, the size of the largest carbon reservoir within the Earth system (e.g. Berner, 1980; Soetaert et al., 2000; Aller, 2014; Hülse et al., 2017; LaRowe et al., 2020a). Understanding, quantifying and predicting organic matter degradation and associated biogeochemical dynamics in marine sediments is thus critical for our ability to understand past, present and future biogeochemical cycles and

climate dynamics.

Benthic biogeochemical dynamics are driven by a complex and dynamic interplay of transport and reaction processes that operate over different temporal and spatial scales. This underlying complexity compromises our ability to understand, quantify and predict diagenetic dynamics. In this respect, reaction-transport models (RTMs) that account for transport (advection, molecular diffusion, bioturbation, and bioirrigation) and reaction (production and consumption) processes, are, in combina-

tion with observational data, powerful analytic, diagnostic and predictive tools. Because of its prominent role, the description and parametrization of OM degradation in diagenetic models is critical for their ability to capture and predict sediment-water exchange and burial fluxes. The rate of OM degradation and, thus, the intensity of the associated biogeochemical cycling is first and foremost driven by the supply of OM at the sediment-water interface. However, it is not only the quantity of that flux that determines benthic biogeochemical rates, but also its quality (Middelburg et al., 1993; Arndt et al., 2013; LaRowe et al.,

2020b). In particular, the benthic redox zonation and benthic recycling fluxes are largely driven by the susceptibility of OM to microbial degradation (Freitas et al., 2020).

Existing OM degradation models can be broadly divided into two different classes according to their description of apparent reactivity of the bulk OM, $k$(t), and its evolution over time which depends on a complex and dynamic interplay of many different factors (Arndt et al., 2013). Discrete models, also called multi-G models, divide the bulk OM into several discrete

fractions, $OM_i$, each characterized by a different reactivity, $k_i$ (Berner, 1980; Jørgensen, 1978). The apparent reactivity of the bulk OM, $k$(t), and its evolution over time is then given by the weighted sum of the individual reactivities $k_i$:

$$k(t) = \sum_i k_i \cdot f_i(t), \text{ with } f_i(t) = \frac{OM_i(t)}{\Sigma_i OM_i(t)}. \tag{1}$$

OM is degraded according to first-order degradation kinetics and the change in OM concentration with burial time/depth is

thus given by:

$$\frac{dOM}{dt} = -k(t) \cdot OM(t) = \sum_i -k_i \cdot OM_i(t). \tag{2}$$





Continuum models, on the other hand, assume a continuous distribution of OM compounds over the reactivity spectrum, thus avoiding the need to partition the bulk material into a limited number of discrete compound classes. One can distinguish theoretically derived continuum models (Boudreau and Ruddick, 1991) and empirical derived power laws (Middelburg, 1989) that are mathematically equivalent (Tarutis, 1993; Jørgensen, 1978; Middelburg, 1989). Although the initial distribution of

OM compounds cannot be inferred from observations, widely used continuum models, such as the reactive continuum model (RCM) apply a gamma distribution because of its mathematical flexibility and its ability to capture observed OM degradation dynamics (Boudreau and Ruddick, 1991). Organic matter degradation can then also be formulated as a first-order degradation rate law with an apparent OM reactivity that decreases with burial time/depth, $k(t)$, (Boudreau and Ruddick, 1991):

$$\frac{dOM}{dt} = -k(t) \cdot OM(t) \tag{3}$$

with:

$$k(t) = \frac{\nu}{a+t}, \tag{4}$$

where $a$ and $\nu$ are free, positive parameters that control the shape of the initial distribution of OM compounds over the reactivity spectrum and, thus the evolution of reactivity with burial time/depth.

The application of continuum models in the framework of local, one-dimensional diagenetic models is straight forward as these models generally solve the coupled reaction-transport equations numerically. However, computational power rapidly becomes a limiting factor if a coupled set of reaction-transport equations were to be numerically solved on a global scale and over longer timescales. Therefore, global diagenetic models designed for large scale applications and the coupling to Earth System Models are generally either highly simplistic, limited in scope or rely on the analytical solution of the reaction-transport

equation (Hülse et al., 2017, 2018). However, finding an analytical solution for complex reaction networks and depth-varying parameters is difficult and requires several simplifying assumptions. For instance, the recently developed one-dimensional and numerically efficient diagenetic model, Organic Matter ENabled SEDiment model (OMEN-SED) assumes steady-state conditions and accounts for depths changes in parameters and reaction processes by dividing the sediment into a set of discrete, yet dynamic biogeochemical zones (Hülse et al., 2018). The considered diagenetic processes are, either expressed as a function

of OM degradation or as zero- or first-order reactions. In the original version of OMEN-SED, bulk OM degradation is described by a multi-G model (i.e. as a sum of first-order degradation rate laws). Consequently, related biogeochemical processes, such as different metabolic pathways, can be expressed as a series of exponential terms (i.e. the analytical solution of the OM conservation equation), for which a general analytical solution can be easily found (Hülse et al., 2018). However, the key limitation of such a multi-G type approach is that it converges to a constant apparent OM reactivity at depth and thus fails to

capture the widely observed continuous decrease in OM reactivity with burial time/depth (Middelburg and Meysman, 2007). Although this limitation can be somewhat mitigated by introducing a larger number of compound classes, each new compound class $i$ requires constraining two new parameters ($k_i$ and $OM_i$). Yet, because of the difficulty associated with identifying discrete classes based on available observation data, the total number of classes that can be reasonably well constrained is





typically restricted to a maximum of three (Jørgensen, 1978; Middelburg, 1989). In addition, due to the lack of a theoretical framework that would allow to constrain OM reactivity on a global scale, the choice of OM degradation model parameters $k_i$ and $OM_i$ is particularly challenging for global scale applications (Hülse et al., 2018). Reported degradation rate constants in marine sediment have been shown to vary by about 10 orders of magnitude (Middelburg et al., 1993; Arndt et al., 2013). Their

spatial variability on the global scale, as well as their response to changing environmental conditions over, for instance, past extreme climate events or to projected climate change is largely unknown. Furthermore, the timescales of OM degradation in the water column (days to weeks) are vastly different than those found in marine sediments ($10 - 10^6$ years). Consequentially, model parameters of OM degradation cannot be directly inferred from the pelagic parametrization. The selected, discrete OM pools of the pelagic model are optimized for the representation of OM degradation dynamics on short-time scales and can thus

not be applied to describe the slower, long-term OM degradation dynamics in the underlying sediment (Hülse et al., 2018).

Continuum models, on the other hand, merely require constraining the two free parameters that define the shape of the initial distribution and the continuous decrease of OM reactivity with time/depth. Therefore, the RCM approach captures a wider range of OM reactivity scenarios over both short and long timescales, while using fewer parameters than multi-G models (Middelburg, 1989; Jørgensen, 1978). In addition, the RCM is in better agreement with our theoretical and qualitative

understanding of the OM degradation process (Aller and Blair, 2006). The application of a RCM approach in a global diagenetic model designed for large scale applications and the coupling to Earth System Models would ease model parametrization requirements. Yet, the RCM cannot be directly applied in OMEN-SED since OMEN-SED's analytical solution relies on the exponential decrease of OM concentration (see Hülse et al., 2018, for details).

Therefore, we here developed a multi-G approximation of the RCM and directly integrated it into the mathematical frame-

work of OMEN-SED. We first provide a short summary of OMEN-SED, including a description of the general model approach and the generic algorithm used to match internal boundary conditions and to determine the integration constants for the analytical solutions. We provide a detailed description of the newly integrated RCM approximation for organic matter degradation, followed by an evaluation of the performance of OMEN-SED-RCM by 1) comparing simulated depth profiles of OM, terminal electron acceptors (TEAs) and metabolic by-products with observations from selected sites, as well as 2) benthic exchange

fluxes of TEAs along a global ocean depth transect. In addition to the model-data comparison, OMEN-SED-RCM results are also compared with original OMEN-SED simulation results. We then force OMEN-SED-RCM with global observational data sets of sediment surface OM contents and bottom water concentrations to explore global patterns of $O_2$ penetration depths and benthic-pelagic exchange fluxes of $O_2$. Finally, to evaluate the performance of OMEN-SED-RCM in reproducing diagenetic dynamics in the deeper anoxic sediment, we use OMEN-SED-RCM to explore the response of the sulfate-methane transition

zone to changes in OM reactivity (i.e. RCM parameter $a$) and sedimentation rate.




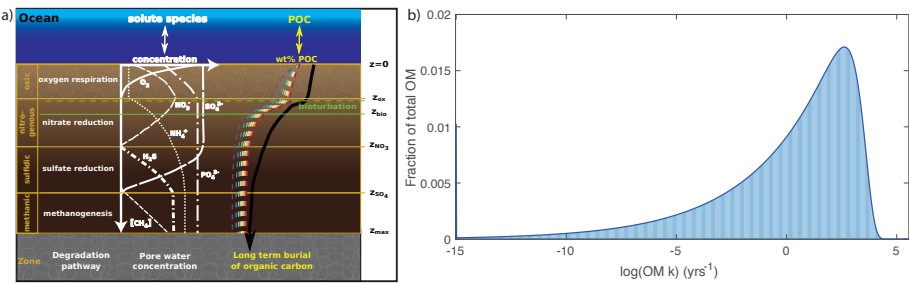

**Figure 1.** a) Illustration of the different modelled species and redox zones resolved in OMEN-SED. Here showing the case $z_{ox} < z_{bio} < z_{NO_3^-} < z_{SO_4^{2-}}$. The black line represents the depth-profile of bulk OM concentration, which is the sum of all OM fractions, $i$ (coloured lines) (adapted from Hülse et al., 2018); b) Approximation of the reactive continuum model (continuous dark blue line) with a 500-G model (light blue bars).



## 2 Model description

OMEN-SED-RCM is based on the diagenetic model OMEN-SED. Hülse et al. (2018) provides a detailed description and evaluation of OMEN-SED. Briefly, OMEN-SED simulates benthic uptake, recycling and burial fluxes based on the vertically resolved conservation equation for solid and dissolved species in porous media (e.g. Berner, 1980; Boudreau, 1997):

$$\frac{\partial \xi C_i}{\partial t} = -\frac{\partial}{\partial z}\left(-\xi D_i \frac{\partial C_i}{\partial z} + \xi w C_i\right) + \xi \sum_j R_i^j \tag{5}$$

where $C_i$ is the concentration of biogeochemical species $i$, $\xi$ equals the porosity $\phi$ for solute species (which is defined as $\phi = \frac{\text{volume porewater}}{\text{volume porewater + solid sediment}}$) and $(1-\phi)$ for solid species. The term $z$ denotes the sediment depth, $t$ is the time, $w$ is the sedimentation rate, $D_i$ is the apparent diffusion coefficient of dissolved species $i$, and $\sum_j R_i^j$ represents the sum of all biogeochemical rates $j$ affecting species $i$.

OMEN-SED accounts for both the advective and diffusive transport of dissolved and solid species. Sedimentation is simulated as a constant rate, $w$, and neglects the effect of sediment compaction (i.e. $\frac{d\phi}{dz} = 0$) due to mathematical constraints (see Hülse et al., 2018). Fick's law of molecular diffusion describes the transport of dissolved species with a species-specific apparent diffusion coefficient $D_{\text{mol,i}}$. The effect of bioturbation is parametrized by a diffusive term (Boudreau, 1996) with a constant bioturbation coefficient $D_{\text{bio}}$ within the bioturbated zone ($z \leq z_{bio}$). The reaction network implemented in OMEN-SED accounts for the most important primary and secondary redox reactions, equilibrium reactions, mineral dissolution and precipitation. Adsorption and desorption processes associated with OM degradation dynamics that affect dissolved and solid species are also explicitly resolved in the model. The resulting set of non-linear, depth-varying coupled equations can per se only be solved numerically. To find an analytic solution, OMEN-SED assumes steady-state conditions and expresses all explicitly resolved biogeochemical processes as a function of OM degradation, as zero- or first-order reactions. In addition, it divides the sediment into a bioturbated ($z < z_{\text{bio}}$) and a non-bioturbated zone ($z_{\text{bio}} \leq z$), as well as an superimposed set of discrete, yet dynamic redox zones (Hülse et al., 2018): 1) an oxic zone ($0 < z \leq z_{\text{ox}}$); 2) a denitrification (or nitrogenous) zone ($z_{\text{ox}} < z \leq z_{\text{NO}_3}$); 3) a sulfate reduction zone ($z_{\text{NO}_3} < z \leq z_{\text{SO}_4}$); and finally, 4) a methanogenic zone ($z > z_{\text{SO}_4}$) (Fig. 1). OMEN-SED allows bioturbation to occur in the anoxic zones of the sediment (here all zones $z > z_{\text{ox}}$ combined), and so does not limit the maximum bioturbation depth to be restricted to a specific redox zone. The extent of each redox zone is dynamic, and the depth of each redox boundary varies in response to changes in ocean boundary conditions and forcing. The zones are interlinked by imposing continuity in concentrations and fluxes at the dynamic internal boundaries ($z_b \in z_{\text{bio}}, z_{\text{ox}}, z_{\text{NO}_3}, z_{\text{SO}_4}$, (compare e.g. Ruardij and Van Raaphorst, 1995; Billen, 1982)). The general equation (eq. 5) is applied to each redox zone with depth invariant parameters, but parameters and the formulation of the reaction term ($\sum_j R_i^j$) in Eq. 5 can vary between the redox zones. OMEN-SED only resolves the most pertinent reaction process within each redox zone and, thus, simplifies the mathematical description of the reaction network, while retaining its biogeochemical complexity (Hülse et al., 2018).

Here, we extend the original 2-G-model description of OM degradation incorporated in the original version of OMEN-SED (v 1.0) with a multi-G approximation of the reactive continuum model (RCM). The new version, OMEN-SED-RCM, thus accounts for a continuous, yet dynamic distribution of OM compounds over a range of reactivities and captures the widely





observed, continuous decrease in apparent OM reactivity with depth/burial age (Boudreau and Ruddick, 1991, and see Fig. 1). Within the RCM the rate of organic matter degradation, $R_{OM}$, is given by:

$$R_{OM} = -\int_0^\infty k \cdot om(k,t)dk \qquad (6)$$

where $om(k,t)$ denotes a probability density function that determines the amount of bulk OM characterized by a reactivity between $k$ and $k + \mathrm{d}k$ at time t. The initial distribution of organic compounds, $om(k,0)$, may take different mathematical forms, but cannot be inferred by observations, but a gamma function is often used due to its mathematical properties Boudreau and Ruddick (1991), following Aris (1968) and Ho and Aris (1987). Assuming first order degradation kinetics, the initial (t = 0) distribution of $om$ over $k$ is given by:

$$om(k,0) = \frac{OM(0) \cdot a^\nu \cdot k^{\nu-1} \cdot e^{-a \cdot k}}{\Gamma(\nu)} \qquad (7)$$

where $OM(0)$ is the initial OM content (at the sediment water interface, SWI), $\Gamma$ is the gamma function, $a$ (yr) is the average lifetime of the more reactive components of the mixture and $\nu$ is a dimensionless parameter determining the shape of the distribution near $k = 0$. The free parameters $a$ and $\nu$ completely determine the shape of the initial OM compound

distribution over the range of $k$. They thus control the overall reactivity of the bulk OM and its evolution with depth/burial age. High $\nu$ and low $a$ values indicate the presence of highly reactive compounds that get rapidly degraded, leading to a rapid decrease in OM reactivity with depth/time. Low $\nu$ and high $a$ values, on the other hand, indicate a dominance of less reactive compounds within the initial distribution, resulting in a lower bulk OM reactivity and a slower decrease in reactivity with depth. Although the choice of the gamma function that describes the initial distribution of OM compounds is partly guided by

mathematical experience, it has been shown to capture the widely observed dynamics of OM degradation in marine sediments well (Arndt et al., 2013). Assuming steady-state conditions (i.e. $\frac{\partial OM}{\partial t} = 0$), the depth profile of bulk OM concentration is given by Boudreau and Ruddick (1991):

$$OM(z) = OM(0) \cdot k(\nu,a) = OM(0) \cdot \left(\frac{a}{a+t(z)}\right)^\nu \qquad (8)$$

where $OM(0)$ is the known concentration at the SWI and $t(z)$ refers to the age of the sediment layer at depth $z$.

Because the OMEN-SED framework relies on an exponential decrease of OM concentration with depth and requires invariant first-order degradation rate constants, we here approximate the reactive continuum model by a multi-G description of OM degradation (see Fig. 1b). In other words, bulk organic matter is represented by large number of $i$ distinct compound classes, $OM_i$, each degraded according to a first-order degradation rate constant, $k_i$. The initial fraction of organic matter in compound

class $i$, $OM_i$, and its respective degradation rate constant, $k_i$, can be calculated based on the initial probability density function by determining the amount of OM within a given reactivity range $k$ and $k + \mathrm{d}k$ at t = 0 (Eq. 6). Because the initial probability





density function is fully described by the two free parameters $a$ and $\nu$, OMEN-SED-RCM only requires defining two parameters instead of the $2\cot i$ parameters of the multi-G description.

Based on Eq. 7, the initial amount of OM within a discrete reactivity class $k$ can be calculated as:

$$f(k,0) = \frac{om(k,0)}{OM(0)} = \frac{a^\nu \cdot k^{\nu-1} \cdot e^{-a \cdot k}}{\Gamma(\nu)} \tag{9}$$

The initial fraction of OM within the reactivity range between 0 and $k$, i.e. having a reactivity $\leq k$ at $t = 0$, is then given by integrating Eq. 9, assuming $a$, $\nu$, $k > 0$:

$$F(k,0) = \int_0^k \left( \frac{om(k,0)}{OM(0)} \right) dk = \frac{a^\nu}{\Gamma(\nu)} \int_0^k \left( k^{\nu-1} \cdot e^{-a \cdot k} \right) dk = \frac{a^\nu}{\Gamma(\nu)} \left[ -\frac{\Gamma(\nu, a \cdot k) \cdot k^\nu}{(a \cdot k)^\nu} \right]_0^k =$$

$$\left[ -\frac{\Gamma(\nu, a \cdot k)}{\Gamma(\nu)} \right]_0^k = -\frac{\Gamma(\nu, a \cdot k)}{\Gamma(\nu)} + \frac{\Gamma(\nu, 0)}{\Gamma(\nu)} = 1 - \frac{\Gamma(\nu, a \cdot k)}{\Gamma(\nu)} \tag{10}$$

where $\Gamma(\nu, a \cdot k)$ denotes the inverse gamma function and $\Gamma(\nu, 0) = \Gamma(\nu)$. To ensure a full coverage of the relevant reactivity space, we recommend choosing a reactivity range from $k_{min} = 10^{-15}$ to $k_{max} = 10^{-log(a)+2}$, although the lower k limit can be adapted to the timescales resolved. In addition, the total reactivity range should be divided into a least $i=100$ equal reactivity bins to ensure an appropriate approximation of the initial OM distribution in case parameter $a \ll 0.01$ (Fig. 2). However, because computational cost is not a limiting factor for OMEN-SED-RCM, a larger number of classes can also be applied. Here, the RCM is approximated by dividing the reactivity range $k = [10^{-15}, 10^{-log(a)+2}]$ into $n=500$ equal reactivity bins, each characterized by a different reactivity constant, $k_i$, thus ensuring a comprehensive approximation of the gamma function defined by the respective $a$ and $\nu$ values.

The reactivity bins are defined by $k_i$ and $k_i+dk_i$ at $t(0)$:

$$F_i = F(k_i, 0) - F(k_{i-1}, 0) \tag{11}$$

The least and most reactive fraction, $F_{min}$ with $k_{min} = 10^{-15}$ yr$^{-1}$ and $F_{max}$ with $k_{max} = 10^{-log(a)+2}$ yr$^{-1}$, are calculated based on the lower and upper incomplete gamma function, respectively (LaRowe et al., 2020a):

$$F_{min} = \int_0^{k_{min}} f(k_{min}, 0) dk = \frac{\Gamma(\nu, a \cdot k_{min})}{\Gamma(\nu)} \tag{12}$$

$$F_{max} = \int_{k_{max}}^\infty f(k_{max}, 0) dk = \frac{\Gamma(\nu, a \cdot k_{max})}{\Gamma(\nu)}. \tag{13}$$




The derived degradation rate constants for $OM_i$, $k_i$, are then used in the first-order reaction term within the conservation equation for organic matter dynamics (see Hülse et al. (2018) for details about the analytical solution):

$$\frac{\partial OM_i}{\partial t} = 0 = D_{OM_i} \cdot \frac{\partial^2 OM_i}{\partial z^2} - \omega \frac{\partial OM_i}{\partial z} - k_i \cdot OM_i \tag{14}$$

5 with $D_{OM_i} = D_{bio}$ for $z \leq z_{bio}$ and $D_{OM_i} = 0$ for $z > z_{bio}$ and $OM_i = F_i \cdot OM(0)$.

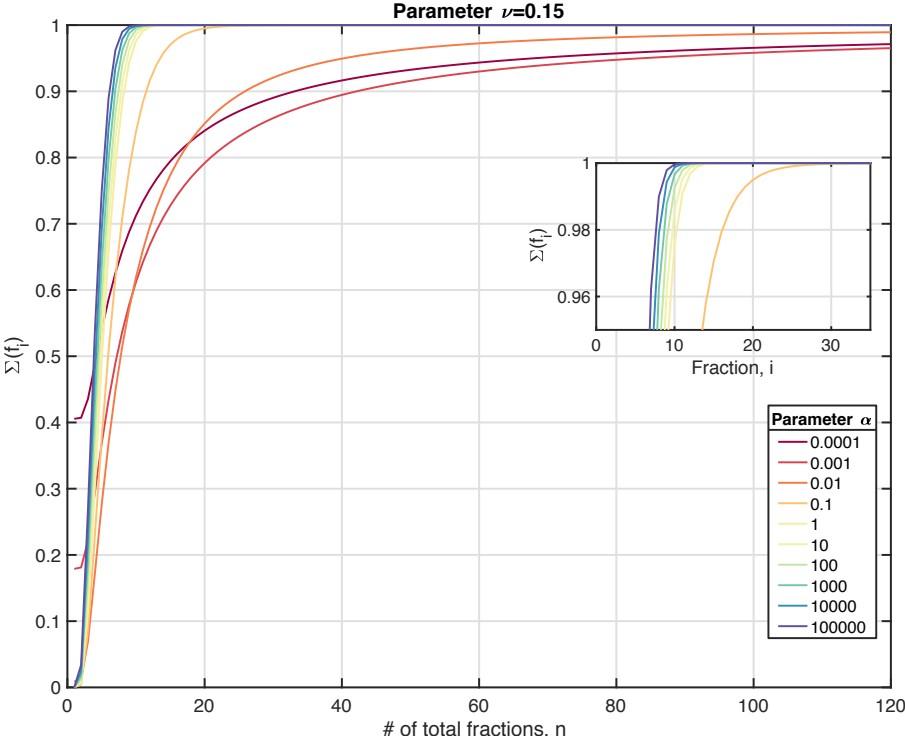

**Figure 2.** Quality of the nG approximation (expressed as the sum of the reactive fractions) as a function of the number of reactivity bins, n, for different RCM parameter $a$ (here shown for $\nu = 0.15$ ). The sum of $\sum_i(f_i)$ should approach 1.0 for an adequate approximation. More reactive OM mixtures (i.e. represented by a small parameter $a$) require a larger number of fractions n than more unreactive OM mixtures (i.e. a larger parameter $a$) to achieve a good approximation of the Gamma function.





## 3 Results and Discussion

### 3.1 Case Study: Sediment Core simulations

**Methodology**

To illustrate the capabilities of OMEN-SED-RCM in simulating local diagenetic dynamics and compare its performance with

the original 2G-model description implemented in OMEN-SED, we here simulate the simulations for the same four sediment cores that were used to benchmark OMEN-SED (Hülse et al., 2018): Santa Barbara Basin (Reimers et al., 1996), Iberian margin and the Nazaré Canyon (Epping et al., 2002). These simulations cover a wide range of different benthic environments reaching from the shallow shelf to the deep-sea, including the highly productive eastern upwelling area on the Iberian Shelf (108 m), the upper slope, organic-rich and non-bioturbated sediments underlying anoxic bottom waters in the Santa Barbara Basin (585

m) and lower slope on the Iberian margin (2213 m), as well as the deep-sea sediments in the Nazaré Canyon (4298 m). For a detailed description of the study sites, see Reimers et al. (1996) and Epping et al. (2002). The respective boundary conditions used in the simulation are provided in Table 1. Additional sediment characteristics (e.g. sedimentation rate, porosity, density), stoichiometric factors and secondary reaction parameters are chosen as in Hülse et al. (2018) (see Tables B1 and C1). The free RCM parameters $a$ and $\nu$ are the optimized to find the best fit between observations and model results, while all other

parameters are kept at their standard value (Hülse et al., 2018).

**Results**

Fig. 3 shows the comparison of simulation results using OMEN-SED-RCM and the original 2G OM model with observed pore water and solid phase profiles for each of the four case study sites. Results show that, despite the underlying simplifications and the reduced fitting effort, OMEN-SED-RCM captures the main observed diagenetic features across the range of different

depositional environments.

More specifically, for the two Iberian margin sites (108 m and 2213 m) OMEN-SED-RCM fits observations generally well. A slight overestimation of $NH_4^+$ production is simulated for the shallower site. A good agreement between simulation results and observations is also found for the Nazaré Canyon station (4298 m), although simulation results underestimate $NH_4^+$ concentrations. The highly variable observed OM depth profile that results from episodic pulses of OM deposition through

the canyon could be equally well fit by several different $a$-$\nu$ combinations, none of these combinations produces a better fit of the $NH_4^+$ profile. Hülse et al. (2018) invoked a number of factors that compromise the model-data fit of the $NH_4^+$ profile at this site. First, Epping et al. (2002), who also did not obtain a better fit with the diagenetic model OMEXDIA, suggest intense bioirrigation activity as a potential driver of higher $NH_4^+$ concentrations. Due to mathematical constrains, OMEN-SED merely includes a simplified description of bioirrigation that would require site specific fitting to account for locally enhanced

irrigation rates. In addition, Hülse et al. (2018) showed that the model data fit can be improved by choosing a different NC-ratio and, thus, reflecting observed variations in Redfield stoichiometries at the Iberian margin and at the Nazaré Canyon sites (Epping et al., 2002).





Overall, a comparison between OMEN-SED and OMEN-SED-RCM results shows that both models reproduce observations equally well. They mainly differ in simulating the deeper parts of the $NH_4^+$ profiles. These differences can be directly linked to the different descriptions of the OM degradation dynamics. While the 2G-model formulation (i.e. OMEN-SED) results in an abrupt decrease in OM degradation rates once the more reactive fraction is consumed, however the RCM (i.e. OMEN-SED-RCM) accounts for the continuous decrease in OM degradation, thus resulting in a continuous consumption of $SO_4^{2-}$ and production of $NH_4^-$ in the first layers under the SWI.

Finally, inversely fitted OM reactivity parameters $a$ and $\nu$ reveal a dominance of highly reactive to reactive OM compounds in the bulk OM across all sites. The consumption of these reactive compounds in the upper sediment layers results in a rapid decrease in apparent OM reactivity with depth for all sites (see Tables C1). These results are not only fully consistent with the previously determined 2G-Model parameters (Hülse et al., 2018), but also explain the good performance of the 2G Model and the small differences between the RCM and 2G Model for these sites. The observed initially high, but rapidly decreasing OM reactivity can be, in contrast to a more slowly decreasing OM reactivity, adequately reproduced by a 2G-Model description.

**Table 1.** Model boundary conditions for the simulated sediment profiles in the Santa Barbara basin (108 and 2213 m) and the Iberian margin (585 and 4298 m) reported in Fig. 3. A DIC bottom water concentration of 2,400 nmols cm$^{-3}$ is assumed for all sites.

**Sediment characteristics:**

| Depth (m) | Temp. (°C) | $z_{bio}$ (cm) | $D_{bio}$ (cm$^2$yr$^{-1}$) | OM (wt%) | $a$ (yr) | $\nu$ [-] |
|---|---|---|---|---|---|---|
| 108 | 12.50 | 0.15 | 0.02 | 4.5 | 0.5 | 1.5 |
| 585 | 5.85 | 0.01 | 0.02 | 6 | 5.16 | 0.32 |
| 2213 | 3.20 | 10.00 | 0.17 | 1.2 | 0.1 | 0.07 |
| 4298 | 2.50 | 4.20 | 0.18 | 2 | 5.3 | 0.17 |

**Bottom water concentrations of solutes** (all in nmol cm$^{-3}$):

| Depth (m) | $O_2$ | $NO_3^-$ | $SO_4^{2+}$ | $NH_4^+$ | $H_2S$ | $PO_4^{3-}$ | Alkalinity |
|---|---|---|---|---|---|---|---|
| 108 | 210.0 | 9.6 | 28,000 | 0.40 | 0.0 | 0.0 | 2,400 |
| 585 | 10.0 | 25.0 | 28,000 | 0.00 | 0.0 | 50.0 | 2,480 |
| 2213 | 250.0 | 25.0 | 28,000 | 0.60 | 0.0 | 0.0 | 2,400 |
| 4298 | 243.0 | 30.1 | 28,000 | 0.22 | 0.0 | 0.0 | 2,400 |



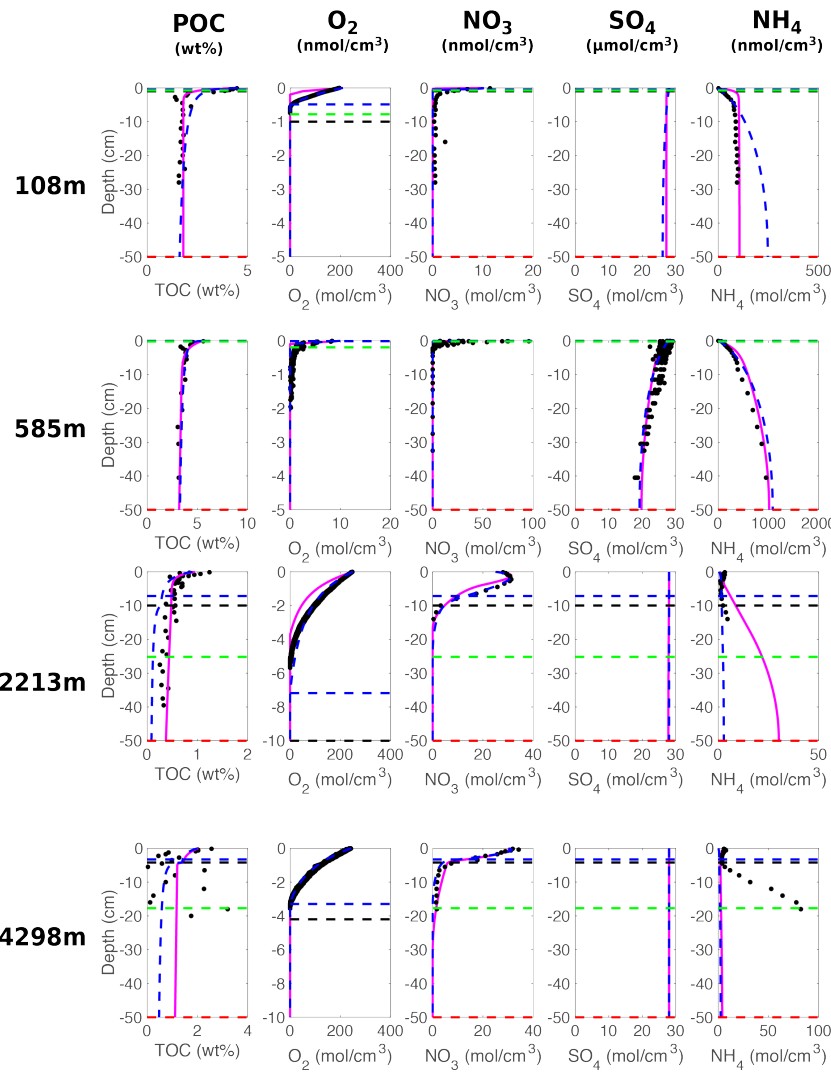

**Figure 3.** OMEN-SED-RCM (blue curves), 2G OMEN-SED (magenta curves) (Hülse et al., 2018) and observed (filled dots) solid phase and dissolved pore water profiles for four different sediment cores. Note that different scales are used for different stations. The horizontal dashed lines in each panel indicate the bioturbation depth (black) and the penetration depths of oxygen (blue), nitrate (green) and sulfate (red) as calculated by the updated OMEN-SED.





## 3.2 Case study: Simulation of Global Ocean Transect

**Methodology**

To evaluate the performance of OMEN-SED-RCM in capturing OM degradation pathways and the resulting TEA-fluxes across different depositional environments we replicate the global ocean transect simulation by Thullner et al. (2009). OMEN-SED-
RCM simulation results are then compared with global observations, as well as the results of OMEN-SED and the complex, numerical diagenetic model used by Thullner et al. (2009). Benthic biogeochemical dynamics and SWI exchange fluxes are simulated for seven sites situated along the global ocean hypsometry (i.e. sea floor depth (SFD) of 100m, 200m, 500m, 1000m, 2000m, 3500m and 5000m). Thullner et al. (2009) used the Biogeochemical Reaction Network Simulator (BRNS, Aguilera et al., 2005), a fully coupled, one-dimensional numerical diagenetic model forced by global observations of bottom water
concentrations and fluxes that are averaged over the respective depth bins. Here, we force OMEN-SED-RCM with the same set of boundary conditions and transport parameters and apply the standard set of OMEN-SED model parameters (Hülse et al., 2018). However, the three applied models differ in their description of OM degradation dynamics. Thullner et al. (2009) and Hülse et al. (2018) apply a simple 1-G model approach, while OMEN-SED-RCM uses the 500G-model approximation of the RCM. Thullner et al. (2009) determined first order degradation rate constants as a function of sedimentation rate for each SFD,
following an empirical relationship proposed by Boudreau (1997). They assume that this rate constant is broadly representative of the mean OM reactivity within the upper, bioturbated 10-20 cm of the sediments. (Hülse et al., 2018) applied the same rate constants in their OMEN-SED run. Here, we constrain the free parameters of the RCM based on a global compilation of previously published values. Inversely determined RCM parameters across a wide range of different environments show that $\nu$ mainly falls between 0.1 - 0.2 (apparent 6th - 11th order of reaction), while parameter $a$ varies globally across orders of
magnitude (Boudreau and Ruddick, 1991; Arndt et al., 2013; Freitas et al., 2020). Based on these previous global results, we thus assume that parameter $a$ controls the global variability in OM reactivity and that parameter $\nu$ remains roughly constant at $\nu = 0.15$ across different depositional environments. Because parameter $a$ is conceptually related to the average lifetime of the more reactive OM components and, thus, the degree of degradation or freshness of OM, the residence time of OM in the water column should exert, among other factors, a control on its magnitude (Boudreau and Ruddick, 1991; Middelburg, 1989).
Boudreau and Ruddick (1991) and (Arndt et al., 2013) found, indeed, a weak trend between parameter $a$ and sedimentation rate:

$$\text{Log}_{10}(a) = 3.35 - 14.81 \cdot \omega, \; r^2 = 0.46. \tag{15}$$

Here, we use the global relationship to estimate parameter $a$ as a function of $\omega$ for sites deeper than 200m water depths along the global hypsometry by using an empirical relationship between $\omega$ and SFD derived from Middelburg et al. (1997) (Table 2).
For high sedimentation rates and thus shallow water depths, the limited available dataset does not fully cover the variability in OM reactivities that results from the dynamic mix of different OM sources in combination with the complex interplay of different controlling factors, such as physical protection mechanisms, microbial community composition or macrobenthic activity (Arndt et al., 2013). As a result eq. 15 tends to underestimate OM reactivity in the coastal ocean. Therefore, we





here assume that the shallowest site (SFD<200m) has an apparent OM reactivity that is characteristic for fresh phytoplankton (Boudreau and Ruddick, 1991). As a result, apparent OM reactivities along the global ocean transect decreases over six orders of magnitude from the shallowest to the deepest SFD: $k = 53.09$ yr$^{-1}$ ($a = 2.8 \cdot 10^{-3}$ yr) at 100m to $k = 7.45 \; 10e^{-5}$ yr$^{-1}$ ($a = 2 \cdot 10^{3}$ yr) at 5000m. This range is like the global range reported in Arndt et al. (2013). Albeit not directly comparable, due to the

different parametrisation of the OM degradation rate, Thullner et al. (2009) assumed a decrease in the overall reactivity over the same depth transect of only one order of magnitude. The parametrisation used in this case is also a function of sedimentation rate:

$$k = 0.38 \cdot \omega(SFD)^{0.59}. \tag{16}$$

In addition to OM reactivity, the reoxidation of reduced substances (i.e. $\gamma_{NH_4}$, $\gamma_{H_2S}$) substantially affects the simulated benthic

fluxes across the SWI. Here we assume that a fixed fraction of $NH_4$ and $H_2S$ is oxidised (i.e. $\gamma_{NH_4} = \gamma_{H_2S} = 0.95$). We acknowledge that choosing a different fraction might change the simulated fluxes, but the resulting range is narrower than the one shown by Hülse et al. (2018) using the 1G OMEN-SED version (see Fig. 4).

**Results**

Fig. 4 compares the results of the simulated benthic TEA fluxes (i.e. $O_2$, $NO_3^{2-}$ and $SO_4^{2-}$) along the global hypsometry using

OMEN-SED-RCM (blue lines) with the original 1-G OMEN-SED version (black lines), the simulation results of Thullner et al. (2009) (red lines) and global observations (black dots/squares/diamonds). Model results are compared with observed $O_2$ and $NO_3^{2-}$ compiled by Middelburg et al. (1997) and $SO_4^{2-}$ compiled by Egger et al. (2018). In general, TEA fluxes simulated with OMEN-SED-RCM are lower than OMEN-SED (Hülse et al., 2018) and BRNS (Thullner et al., 2009) simulations (Fig. 4). This difference can be directly attributed to the different representations of OM degradation dynamics in the three models (i.e.

numerical 1-G, analytical 1-G and analytical 500-G model RCM approximation). The 1-G model applied by Thullner et al. (2009) and Hülse et al. (2018) assumes a constant OM reactivity for the entire sediment column and, therefore, overestimates bulk OM reactivity in the deeper, often anoxic sediment layers, resulting in higher TEA uptake. The RCM approximation on the other hand, accounts for the entire reactivity continuum, and thus captures the decrease in apparent OM reactivity with sediment depth. As a result, OM degradation rates decrease faster with sediment depth, resulting in a reduced TEA uptake.

Fig. 4 shows that OMEN-SED-RCM captures the observed decrease in TEA uptake fluxes from the shallow coastal ocean to the deep abyss well. Like the other models, it slightly underestimates observed deep ocean $O_2$ fluxes (> 2000m). These low $O_2$ fluxes are a direct consequence of the low OM reactivities (i.e. high parameter $a$) estimated for these water depths in combination with a potential bias in $O_2$ uptake flux observations towards more dynamic deep-sea sites that receive by more reactive OM. Eq. 15 likely underestimates OM reactivity for the more dynamic deep-sea sites (Arndt et al., 2013), resulting in

a slight underestimation of fluxes. Furthermore, OMEN-SED-RCM reproduces the observed water depth trend in $NO_3^-$ uptake fluxes well, but, like the other models, underestimates the high $NO_3^-$ uptake fluxes observed in the shallow Pacific Ocean (Middelburg et al., 1996). As suggested by Hülse et al. (2018), an improved model-data fit (green line) can be obtained by applying boundary conditions ($O_2 = 10$ nmol cm$^{-3}$ and $NO_3^- = 80$ nmol cm$^{-3}$) and a C:N elemental ratio (C:N ratio = 0.067)





**Table 2.** Seafloor depth dependency of key model parameters and boundary conditions (adapted from Thullner et al., 2009). Superscripted numbers denote the following references: Derived from [a]Middelburg et al. (1997); [b] Van Cappellen and Wang (1995); [c] Conkright et al. (2002); [d] Boudreau (1997); [e] Calculated with OMEN-SED from OM$_{flux}$; [f] constant value according to Boudreau and Ruddick (1991) and Freitas et al. (2020); [g] see Eq. 4

| | Seafloor depth | | | | | | |
| --- | --- | --- | --- | --- | --- | --- | --- |
| | 100 m | 200 m | 500 m | 1000 m | 2000 m | 3500 m | 5000 m |
| **Model parameters** | | | | | | | |
| $w^a$ [cm yr$^{-1}$] | $3.98 \times 10^{-1}$ | $3.60 \times 10^{-1}$ | $2.67 \times 10^{-1}$ | $1.62 \times 10^{-1}$ | $5.94 \times 10^{-2}$ | $1.32 \times 10^{-2}$ | $2.94 \times 10^{-3}$ |
| $D_{bio}{}^a$ [cm$^2$ yr$^{-1}$] | 27.5 | 25.1 | 19.0 | 12.1 | 4.83 | 1.23 | 0.310 |
| $\phi^b$ [-] | 0.85 | 0.85 | 0.80 | 0.80 | 0.80 | 0.80 | 0.80 |
| T$^c$ [°C] | 10.3 | 9.7 | 8.1 | 5.8 | 3.0 | 1.5 | 1.4 |
| $\rho_{sed}{}^c$ [g cm$^{-3}$] | 2.5 | 2.5 | 2.5 | 2.5 | 2.5 | 2.5 | 2.5 |
| $\nu^f$ [-] | 0.15 | 0.15 | 0.15 | 0.15 | 0.15 | 0.15 | 0.15 |
| $a^g$ [yr] | 0.0028 | 0.0103 | 0.2505 | 9.0317 | 295.72 | 1426.9 | 2025.2 |
| $OMk^h$ [yr$^{-1}$] | 53.09 | 14.4 | 0.58 | 0.016 | 0.00051 | 0.000105 | 7.46e-05 |
| **Upper boundary conditions** | | | | | | | |
| $OM_{flux}{}^a$ [$\mu$mol cm$^{-2}$ yr$^{-1}$] | 510 | 467 | 357 | 228 | 93.0 | 24.3 | 6.33 |
| $OM^e$ [wt%] | 0.79 | 0.78 | 0.55 | 0.50 | 0.42 | 0.32 | 0.25 |
| $O_{2,0}{}^c$ [nmol cm$^{-3}$] | 132 | 129 | 121 | 114 | 116 | 135 | 141 |
| $NO_{3,0}{}^c$ [nmol cm$^{-3}$] | 17.3 | 18.6 | 22.1 | 26.5 | 31.0 | 31.6 | 31.6 |
| $SO_{4,0}{}^b$ [nmol cm$^{-3}$] | 28,000 | 28,000 | 28,000 | 28,000 | 28,000 | 28,000 | 28,000 |





that are more representative of the Eastern Pacific Ocean (see Bohlen et al., 2012). Finally, OMEN-SED-RCM also captures the observed trend in $SO_4^{2-}$ uptake fluxes well. Simulation results also agree well with the OMEN-SED results that assume an almost complete re-oxidation of the deep $H_2S$ flux (i.e. $\gamma_{H2S} = 95\%$ (black line with black circle, hidden by the OMEN-SED-RCM results). In contrast, BRNS model results, as well as OMEN-SED results that assume a weak re-oxidation of the deep

5   $H_2S$ flux (i.e. $\gamma_{H2S} = 0.05\%$ (black line with black triangles) both overestimate anoxic OM degradation and thus $SO_4^{2-}$ uptake flux. Yet, like the reign-specific high $NO_3^-$ uptake fluxes observed in the shallow Pacific Ocean, all model simulations fail to capture locally observed high $SO_4^{2-}$ uptake fluxes of $> 300~\mu mol~cm^{-2}~yr^{-1}$ in the very shallow coastal ocean. Depth-averaged boundary conditions, as well as apparent OM reactivity parameters that are broadly representative for the wider depositional environment cannot reproduce such local coastal dynamics.

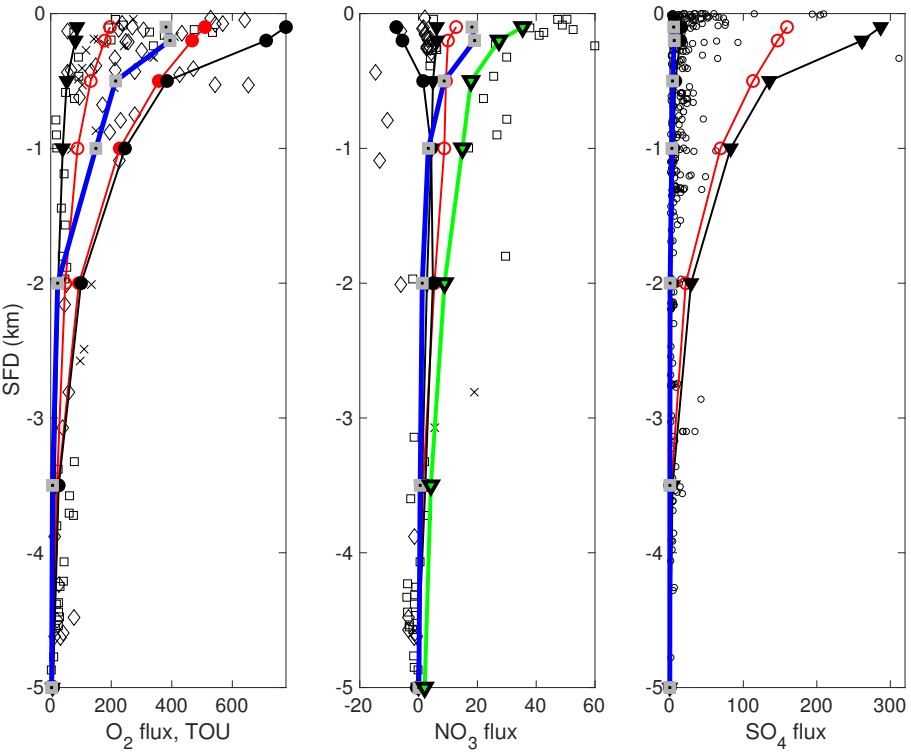

**Figure 4.** Simulated (lines) and observed (points) benthic fluxes of $O_2$, $NO_3^-$ and $SO_4^{2-}$ along a global hypsometry transect. Flux units are in $\mu mol~cm^{-2}~yr^{-1}$. OMEN-SED-RCM (blue lines with grey squares) are compared to: simulation results from the numerical diagenetic model BRNS (red lines with open symbols, Thullner et al. (2009)); and results from the analytical diagenetic model OMEN-SED (black lines, ● : $\gamma_{NH_4} = \gamma_{H_2S} = 0.95$; ▼: $\gamma_{NH_4} = \gamma_{H_2S} = 0.05$), Hülse et al. (2018). The red line with filled circles are total oxygen uptake simulated by Thullner et al. (2009). The green line indicates OMEN-SED-RCM simulations that were forced with boundary conditions and N:C values that are more representative of the Eastern Pacific Ocean. Observed data of $O_2$ and $NO_3^-$ fluxes from the Atlantic (diamonds), Pacific (squares) and Arctic/Indian Ocean(crosses) are from Middelburg et al. (1996), while $SO_4^{2-}$ data (circles) are from Egger et al. (2018).





## 3.3 Global application

### Methodology

To evaluate the performance of OMEN-SED-RCM in reproducing observed global patterns of benthic-pelagic exchange fluxes and sediment redox zonation, we force OMEN-SED-RCM with existing global data sets and parametrizations to simulate

diffusive $O_2$ fluxes (DOU) and oxygen penetration depths (OPD). OM concentrations at the SWI are constrained based on the global $1° \times 1°$ gridded extrapolation of over 5500 observations provided by Seiter et al. (2004). The boundary concentrations for dissolved species ($O_2$, $NO_3^-$, $SO_4^{2-}$, $PO_4^{3-}$) and temperature at the SWI are constrained based on regionally averaged observational data extracted from the World Ocean Atlas (Boyer et al., 2009). The variables are reported with a $1° \times 1°$ horizontal resolution and we assume the last vertical cell at every grid-coordinate represents bottom water solute concentrations. The initial

distribution of organic matter compounds across the reactivity spectrum at the SWI (i.e. parameters $a$ and $\nu$) and, therefore, the apparent reactivity of the depositing OM is constrained based on a compilation of inversely determined RCM parameters across a wide range of different depositional environments (Boudreau and Ruddick, 1991; Arndt et al., 2013; Freitas et al., 2020). Previous studies of inverse models showed that the global variability in apparent OM reactivity is mostly controlled by variations in parameter $a$, while parameter $\nu$ remains comparably constant across environments (Boudreau and Ruddick, 1991;

Freitas et al., 2020). Building on these findings, we thus assume a globally constant parameter $\nu$ value of 0.15. In contrast, parameter $a$ changes globally. Conscious of the risk to over-parametrize the model, we still use the weak trend in parameter $a$ and sedimentation rate found in Arndt et al. (2013), which follows roughly the observed broad global decrease in apparent OM reactivity from coastal to slope and abyssal depositional environments (Arndt et al., 2013).

$$log_{10}(a) = 3.35 - 14.81 \times \omega, \; r^2 = 0.46 \tag{17}$$

Additionally, the lack of estimates of parameter $a$ in shallow environments makes this $a$-$\omega$ trend especially weak in such areas. However, our global boundary condition on OM concentration is limited to depth > 1000m, hence remaining within the limits of the trend. We assume the $a$ values implicitly accounts for broad patterns in benthic environmental characteristics, such as, among others: OM sources, transport pathways and transit times, oxygen exposure, mineral protection, microbial community structure etc. This approach thus captures the broad, generally observed reactivity trend across broadly defined depositional

environments without over-parametrizing the problem. Finally, model transport and all additional reaction rate parameters are chosen according to the original OMEN-SED set-up and summarized in Tables B1 and C1.

### Results

Fig. 5 illustrates the global distribution of simulated DOU and OPD. Simulated global patterns in $O_2$ uptake broadly agree with global DOU observations and their global extrapolation (see Seiter et al., 2005). In addition, the simulated OPDs generally

agree well with observations and capture also the widely observed increase in OPD with water depths (see Fig. 6). The highest $O_2$ uptake rates ($\sim 38$ mmol m$^{-2}$ yr$^{-1}$) and the shallowest OPD are simulated for sediments underlying the western continental margins, the Arabian Sea, parts of the Greenland Sea and Laptev Sea and the equatorial Pacific (Fig. 5 a).





These regions are characterized by the highest apparent OM reactivity, as well as high OM deposition fluxes (Seiter et al., 2005). In these regions, high apparent OM reactivity has been linked to either the efficient transport of large amounts of fresh, marine OM in the form of seasonal pulses or the presence of extended oxygen minimum zones. Both environmental drivers reduce the degree of pelagic OM degradation. Ecosystems that are characterized by a strong seasonality often deliver OM

in strong post-bloom seasonal pulses to the benthic environment (Cavan et al., 2017; Cowie, 2005; Vandewiele et al., 2009; Cowie et al., 1999; Suthhof et al., 2001; Keil et al., 2016). These pulses favour the formation of fast sinking phytodetrital aggregates that reduce the residence time of sinking OM in the water column and, thus, the degree pelagic degradation of OM. Pronounced oxygen minimum zones on the other hand reduce the exposure of OM to oxygen (Fischer et al., 2009; Zonneveld et al., 2010; Vandewiele et al., 2009). In both types of oceanic region, the resulting strong coupling between the euphotic zone

and the sediment supports high oxygen uptake rates and shallow OPD. In agreement with global observations, OMEN-SED-RCM results reveal that, in particular, the eastern part of the equatorial Pacific reveals high oxygen fluxes into the sediment (>4 mmol m$^{-2}$ yr$^{-1}$). However, oxygen uptake decreases westwards, as well as north and southwards due to the rapid decrease in OM flux away from the equatorial upwelling area (Seiter et al., 2005; Smith et al., 1997). In the western equatorial Pacific, higher DOU rates are supported by the increased concentrations of OM close to the active margins in southeast Asia, where

high riverine loads deliver OM to marine sediments. High oxygen uptake and shallow OPD are also observed in sediments underlying eastern boundary currents. However, here, OM reactivities are slightly lower due to the increased input of terrestrial material and strong lateral OM fluxes that redistribute pre-aged OM across the wide shelf of the passive Atlantic margin (Alonso-González et al., 2009; Mollenhauer et al., 2007, 2003; Arthur et al., 1998; Inthorn et al., 2006; Lovecchio et al., 2018) as well as on the narrow shelf of the active margins of the Pacific Ocean (Muñoz et al., 2004; Venkatesan and Kaplan, 1992).

In agreement with global observations, generally low $O_2$ fluxes and deeper OPD are simulated for deep open ocean sediments, like the north (DOU rates < 1 mmol m$^{-2}$ yr$^{-1}$) and south Pacific (DOU rates < 2 mmol m$^{-2}$ yr$^{-1}$) and the Indian Ocean (DOU rates = [0.5 - 3] mmol m$^{-2}$ yr$^{-1}$). Here, both the low magnitude and low quality of the depositing OM limits organic matter degradation and oxygen penetrates deep into the sediment (Glud, 2008; Smith et al., 2001). The lowest $O_2$ uptake rates are generally observed and also simulated for the central, oligotrophic gyre regions of the ocean basins that receive little

OM input. Higher $O_2$ uptake can be observed close to the continental slopes, where lateral OM fluxes support a higher OM deposition (Seiter et al., 2005; Archer and Devol, 1992). Atypically high $O_2$ uptake rates are observed as well as simulated for areas in the deep south Indian Ocean and south Atlantic Ocean. In these areas, seasonal pulses of primary production in the Antarctic polar front support an enhanced deposition of OM to the sediment and, thus support higher benthic oxygen uptake rates (Rabouille et al., 1998).

Furthermore, the reoxidation of reduced products can make up a substantial fraction of the oxygen consumption in marine sediments. These secondary reactions consume products originating from anaerobic mineralization [Aller, 1990; Boudreau and Canfield, 1993] and can make up to 56% (Jørgensen and Kasten, 2006) of the total oxygen consumption in certain environments; thus cannot be ignored. We take these processes into account by running the full version of OMEN-SED-RCM. Additionally, most of the global simulation here occurs in deeper parts of the ocean, where oxygen is the main oxidant of OM.





Finally, OMEN-SED-RCM simulation results also reflect the widely observed inverse relationship between OPD and oxygen uptake rates (Rullkötter, 2006; Glud, 2008; Wenzhöfer and Glud, 2002). Shallow OPDs of less than 10 cm are observed for $O_2$ uptake rates of $\geq 4$ mmol m$^{-2}$ yr$^{-1}$. In contrast, the deep ocean and the central ocean gyres reveal OPDs of several centimetres to metres. Such deep OPD agree with observations from the central South Pacific. Here, D'Hondt et al. (2015) reported OPD that reach down to the basement at $\sim$120 m sediment depth. These deep OPD are directly linked to the extremely low OM deposition, as well as apparent OM reactivities (Røy et al., 2012) that are reflected in low, observed DOU rates (D'Hondt et al., 2009, 2015; Røy et al., 2012; Fischer et al., 2009).





**Figure 5.** *Top*: RCM parameter $a$ as a function of sedimentation rate, $\omega$, according to Arndt et al. (2013): $log_{10}(a) = 3.35 - 14.81 \times \omega$. *Middle*: Global map of simulated DOU. Positive fluxes are directed into the sediments. *Bottom*: Global map of simulated oxygen penetration depth (OPD). Yellow areas indicate that oxygen flows across the bottom of the model domain (i.e. 10 m). White indicates areas for which surface sediment OM contents are not available i.e. most coastal margins and the south-west Pacific (see Seiter et al., 2005).



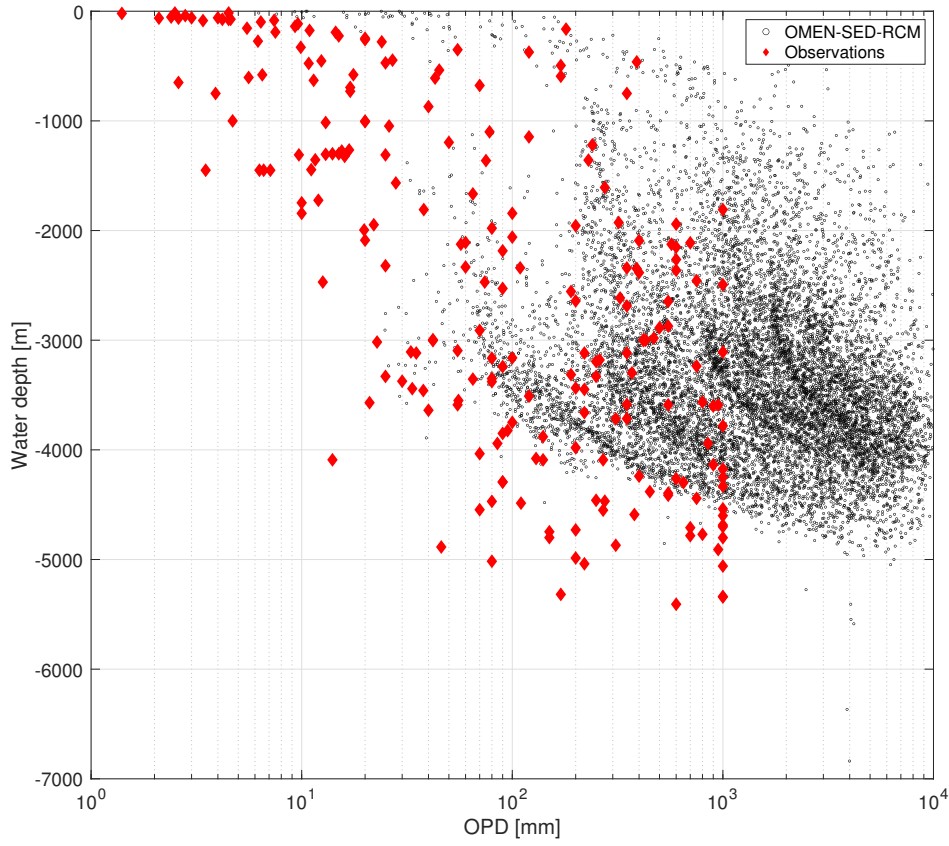

**Figure 6.** OPD versus water depth for observed (red diamond) and simulated (black circles) values. Observations are taken from: Jahnke et al. (1989); Canfield et al. (1993); Glud et al. (1994, 1998, 1999, 2003, 2009); Wenzhöfer et al. (2001a, b); Black et al. (2001); Giordani et al. (2002); Wenzhöfer and Glud (2002); Lansard et al. (2008, 2009); Sachs et al. (2009); Witte et al. (2003a, b); Egger et al. (2018)



## 3.4 Depth of the sulfate-methane transition (SMTZ)

**Methodology**

Sulphate is the most important terminal electron acceptor in marine sediments on a global scale (Jørgensen, 1982; Thullner et al., 2009). Its depth profile is a macroscopic manifestation of long-term OM degradation in the deeper anoxic sediment
5 and, thus, crucial for understanding benthic biogeochemical dynamics. Sulfate depth profiles typically show a down core decrease where the gradient is primarily controlled by OM degradation rates and thus both the amount and quality of OM in the deeper sediment. The sediment depth at which downward diffusing sulphate is completely consumed is referred to the sulphate-methane transition zone (SMTZ). It separates the sulphate reduction from the methanogenic zone. The SMTZ is a very dynamic redox boundary and its position is very sensitive to changes in environmental conditions (Regnier et al., 2011;
10 Meister et al., 2013). Here, we use OMEN-SED-RCM to simulate changes in the position of the SMTZ in response to a range of sedimentation rates and OM reactivity parameters and compare the simulated patterns with previously observed/simulated patterns (Regnier et al., 2011; Meister et al., 2013).

To this end we set-up a one-dimensional OMEN-SED-RCM simulation that is representative for a shallow depositional environment with a high OM input and a high sedimentation rate. OM concentration is set to 1.1 wt%. We assume bottom
15 water concentrations of $SO_4^{2-}$ = 28 mM and $CH_4$ = 0 mM at the SWI. The model domain is set to 45 m and thus accounts for the active part of the anoxic sediment layer. We use this OMEN-SED-RCM set-up to explore the response of the SMTZ to variations in sedimentation rates and apparent OM reactivity (i.e. parameter $a$). An overview of the applied boundary conditions and parameters is given in Table 3. Because, we do not consider active methane venting or upward fluid flow, the location of the SMTZ is mainly controlled by OM degradation.

**Table 3.** Model boundary conditions representative of shallow shelf marine sediments for the SMTZ simulations. Derived from *: (Burwicz et al., 2011)

| **Model parameters:** | | | | | |
|---|---|---|---|---|---|
| TOC | Depth | $\omega^*$ | $O_2$ | $a$ | $\nu$ |
| wt% | [m] | [cm yr$^{-1}$] | [mmol cm$^{-3}$] | [yr] | [-] |
| 1.1 | 50 - 250 | 0.1212 - 0.0456 | 270 | $10^0 - 10^7$ | 0.15 |





## Results

In agreement with results of Regnier et al. (2011) and Meister et al. (2013), OMEN-SED-RCM simulation results show that the sedimentation rate, $\omega$, and parameter $a$ exert a dominant control on the location of the SMTZ (Fig. 7). OMEN-SED-RCM captures the observed/simulated response of the SMTZ to changing environmental conditions well and results are similar to those found in Regnier et al. (2011) and Meister et al. (2013). More specifically, in agreement with previous results, OMEN-SED-RCM shows that very high apparent OM reactivities generally lead to a deep SMTZ. A decrease in apparent OM reactivity (i.e. increase in parameter $a$) then leads to an upward migration of the SMTZ, until it reaches a minimum depths around intermediate OM reactivity. Hereafter, a further decrease in apparent OM reactivity triggers a new down core migration of the SMTZ. This somewhat counter-intuitive behaviour can be explained by a limitation of sulfate reduction rates in the area of both high and low OM reactivities. If apparent OM reactivity is high, a significant fraction of the OM will be degraded in the shallow sediment layer, thus reducing the supply of reactive OM to the sulfate reduction zone. As a consequence, sulfate reduction rates and, thus, $SO_4^{2-}$ consumption become limited by a lack of degradable substrate. On the other hand, if apparent OM reactivity is low, sulfate reduction rates become limited by OM reactivity. Both scenarios result in a deep SMTZ. At intermediate OM reactivities, both the quantity and the quality of the OM that is buried into deeper sediment layers is sufficiently high to support enhanced sulfate reduction rates, thus shifting the SMTZ upwards. OMEN-SED-RCM results show that the location of the SMTZ is substrate limited for values below $\log(a)$ =[1-3] years (depending on sedimentation rates) and becomes reactivity limited above approximately $\log(a)$ = [4.4 - 4.5] years.

Simulation results also show that OM reactivity exerts the dominant control on the depth of the SMTZ. Changes in sedimentation rate and OM deposition shift (not shown here), but do not change the general pattern. Increasing the deposition flux of OM at the SWI would lead merely to the shallowing of the SMTZ on both ends of the parameter $a$ spectrum (not shown here). Increasing sedimentation rates generally result in shallower SMTZ in the area of high OM reactivities because they increase the flux of OM to the deeper sediment layers. For the same reason, high sedimentation rates also decrease the minimum depth of the SMTZ for intermediate reactivities. The shallowest SMTZ (-5 m) is simulated for the highest $\omega$ value and $a = 10^{2.7}$ years, while the deepest at -20m occurs when $\omega$ is lowest and $a = 10^{3.6}$ years. In the area of low reactivities (i.e. high parameter $a$), sedimentation rates, on the other hand, only exert a limited effect on the location of the SMTZ because higher sedimentation rates merely result in a slight increase in OM reactivity at depth.





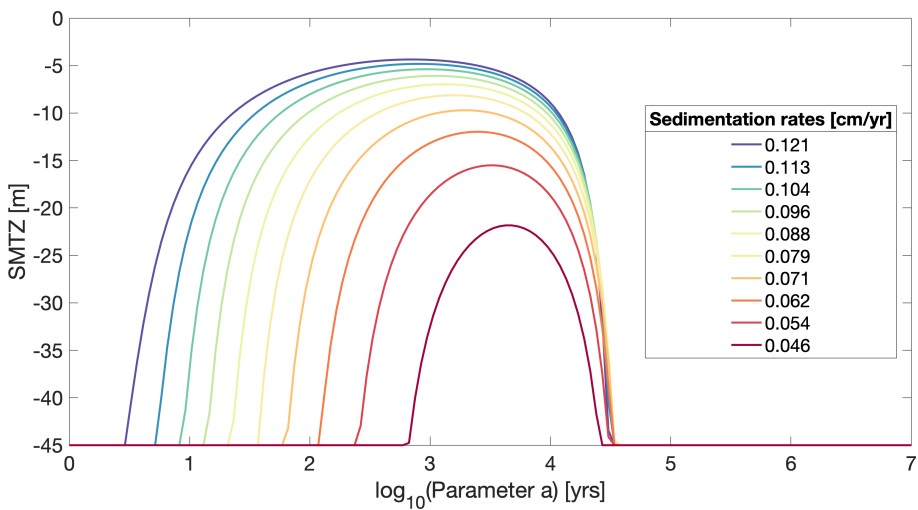

**Figure 7.** Depth of the sulfate-methane transition (SMTZ) as a function of OM reactivity parameters and sedimentation rates ($\omega$). For $\mathrm{Log}_{10}(a)$< 2.9 years and $\mathrm{Log}_{10}(a)$> 4.5 years $\mathrm{SO}_4^{2-}$ always reaches the bottom of the model domain. OM = 1 wt%; $\nu$=0.15.



# 4 Conclusions

Here we present OMEN-SED-RCM, an extension of the original analytical diagenetic model OMEN-SED (Hülse et al., 2018). OMEN-SED-RCM replaces the simple 2G OM degradation model of the original version with a pseudo-RCM representation of OM degradation. Because the analytical solutions of the coupled transport-reaction equations that underlie OMEN-SED
require an exponential form of the OM degradation term, the RCM is approximated by a multi-G approximation, applied here with 500 fractions (Boudreau and Ruddick, 1991; Dale et al., 2015). The RCM and its multi-G approximation (i.e. reaction rate constants and initial fraction of each OM class) is fully constrained by the two free RCM parameters $a$ and $\nu$ that control the initial distribution of OM compounds over the reactivity spectrum.

We show that the new version of OMEN-SED is not only able to reproduce observed pore water profiles across a wide range
of depositional environments and captures observed global patterns of TEA-fluxes, oxygen penetration depths, biogeochemical reaction rates, but also accounts for the widely observed continuous decrease in OM reactivity with sediment depths/burial time and, thus provides a more realistic description of anaerobic degradation pathways. This added functionality offers an alternative to the common, but simpler 2G-model description implemented in the original model, extending the model's applicability to a wider range of environments and timescales, while requiring fewer parameters to describe a wider spectrum of OM reactivity.
These improvements were implemented while maintaining the computational advantages of the original version.

*Code availability.* The commented OMEN-SED-RCM source code (MATLAB) is provided as a Supplement to this article and is also available for download on the web Pika (2021). The specific version of the OMEN-SED model used in this manuscript is tagged as release v1.1, has been assigned a DOI of 10.5281/zenodo.4029488 and is hosted on GitHub. A version including the code to plot some of the presented results has the DOI of 10.5281/zenodo.4421777 and is hosted on GitHub.

*Author contributions.* PP and SA conceived and designed the study. PP: Carried out the model simulations, wrote, validated the new model code, analysed the data and wrote the manuscript. DH: Provided the nutrient data, helped with the implementation and contributed the editing of the manuscript. SA: Contributed the editing of the manuscript.

*Competing interests.* The authors declare that they have no conflict of interest.

*Acknowledgements.* SA and PP were supported by funding from the European Union's Horizon 2020 research and innovation programme
under the Marie Skłodowska-Curie grant agreement no. 643052 (C-CASCADES). D.H. is supported by a postdoctoral fellowship from the Simons Foundation (Award ID 653829).





**Appendix A:   Reaction Network**

**Appendix B:   Model transport parameters**

**Appendix C:   Biogeochemical model parameters**





**Table A1.** Primary pathways of organic matter degradation, secondary redox reactions and stoichiometries implemented in the reaction network.

| Pathway | Stoichiometry |
|---|---|
| | **Primary Redox reactions** |
| Aerobic degradation | $(CH_2O)_x(NH_3)_y(H_3PO_4)_z + (x+2y)O_2 + (y+2z)HCO_3^- \rightarrow (x+y+2z)CO_2 + yNO_3^- + zHPO_4^{2-} + (x+2y+2z)H_2O$ |
| Denitrification | $(CH_2O)_x(NH_3)_y(H_3PO_4)_z + \frac{(4x+3y)}{5}NO_3^- \rightarrow \frac{(2x+4y)}{5}N_2 + \frac{(x-3y+10z)}{5}CO_2 + \frac{(4x+3y-10z)}{5}HCO_3^- + zHPO_4^{2-} + \frac{(3x+6y+10z)}{5}H_2O$ |
| Sulfate reduction | $(CH_2O)_x(NH_3)_y(H_3PO_4)_z + \frac{x}{2}SO_4^{2-} + (y-2z)CO_2 + (y-2z)H_2O \rightarrow \frac{x}{2}H_2S + (x+y-2z)HCO_3^- + yNH_4^+ + zHPO_4^{2-}$ |
| Methanogenesis | $(CH_2O)_x(NH_3)_y(H_3PO_4)_z + (y-2z)H_2O \rightarrow \frac{x}{2}CH_4 + \frac{x-2y+4z}{2}CO_2 + (x-2z)HCO_3^- + yNH_4^+ + zHPO_4^{2-}$ |
| | **Secondary Redox reactions** |
| Nitrification | $NH_4^+ + 2O_2 + 2HCO_3^- \rightarrow NO_3^- + 2CO_2 + 3H_2O$ |
| Sulfide oxidation | $H_2S + 2O_2 + 2HCO_3^- \rightarrow SO_4^{2-} + 2CO_2 + 2H_2O$ |
| AOM | $CH_4 + CO_2 + SO_4^{2-} \rightarrow 2HCO_3^- + H_2S$ |
| | **Adsorption reactions and mineral precipitation** |
| NH$_4$ adsorption | $NH_4^+ \xrightarrow{K_{NH4}} NH_4^+ \text{(ads)}$ |
| P ad-/desorption | $PO_4^{2-} \xrightarrow{K_{PO_4}^{I,II}} PO_4^{2-}\text{(ads)};$ $\qquad PO_4^{2-} \xrightarrow{k_s} Fe-bound\ P \xrightarrow{k_m} PO_4^{2-}$ |
| CFA precipitation | $PO_4^{2-} \xrightarrow{k_a} CFA$ |





**Table B1.** Sediment characteristics and transport parameters.

| Parameter | Unit | Value | Description/Source |
|---|---|---|---|
| $\rho_{sed}$ | g cm$^{-3}$ | 2.6 | Sediment density |
| $w$ | cm yr$^{-1}$ | Fct. of seafloor depth or from ESM | Advection/Sediment accumulation rate (Middelburg et al., 1997) |
| $z_{bio}$ | cm | 10 or 0.01 | Bioturbation depth (Boudreau, 1998; Teal et al., 2010) |
| $D_{bio}$ | cm$^2$ yr$^{-1}$ | Fct. of seafloor depth | Bioturbation coefficient (Middelburg et al., 1997) |
| $\phi$ | - | 0.85 | Porosity |
| F | - | $\frac{1}{\phi^m}$ | Tortuosity, here m=3 |
| f$_{ir}$ | - | 1 | Irrigation factor |
| **Diffusion coefficients** (Li and Gregory, 1974; Schulz, 2006; Gypens et al., 2008) | | | |
| $D^0_{O_2}$ | cm$^2$ yr$^{-1}$ | 348.62 | Molecular diffusion coefficient of oxygen at 0°C |
| $D^T_{O_2}$ | cm$^2$ yr$^{-1}$ °C$^{-1}$ | 14.09 | Diffusion coefficient for linear temp. dependence of oxygen |
| $D^0_{NO_3}$ | cm$^2$ yr$^{-1}$ | 308.42 | Molecular diffusion coefficient of nitrate at 0°C |
| $D^T_{NO_3}$ | cm$^2$ yr$^{-1}$ °C$^{-1}$ | 12.26 | Diffusion coefficient for linear temp. dependence of nitrate |
| $D^0_{NH_4^+}$ | cm$^2$ yr$^{-1}$ | 309.05 | Molecular diffusion coefficient of ammonium at 0°C |
| $D^T_{NH_4^+}$ | cm$^2$ yr$^{-1}$ °C$^{-1}$ | 12.26 | Diffusion coefficient for linear temp. dependence of ammonium |
| $D^0_{SO_4}$ | cm$^2$ yr$^{-1}$ | 157.68 | Molecular diffusion coefficient of sulphate at 0°C |
| $D^T_{SO_4}$ | cm$^2$ yr$^{-1}$ °C$^{-1}$ | 7.88 | Diffusion coefficient for linear temp. dependence of sulfate |

Note: DIC and ALK coefficients are the values of HCO$_3^-$ from Schulz (2006).



**Table C1.** Values for biogeochemical parameters used in OMEN-SED-RCM. The variables $x$, $y$ and $z$ denote the elemental ratio of carbon, nitrogen and phosphorus of the degrading OM (here set to $C : N : P = 106 : 16 : 1$).

| Parameter/Variable | Unit | Value | Description |
|---|---|---|---|
| **Stoichiometric factors and molecular ratios** | | | |
| $N : C_i$ | mol/mol | $\frac{y}{x} = \frac{16}{106}$ | Nitrogen to carbon ratio |
| $P : C_i$ | mol/mol | $\frac{z}{x} = \frac{1}{106}$ | Phosphorus to carbon ratio |
| $M : C$ | mol/mol | 0.5 | Methane to carbon ratio |
| | | | produced during methanogenesis |
| $DIC : C^I$ | mol/mol | 1.0 | DIC to carbon ratio until $z_{SO_4}$ |
| $DIC : C^{II}$ | mol/mol | 0.5 | DIC to carbon ratio below $z_{SO_4}$ |
| $O_2 : C$ | mol/mol | $\frac{x+2y}{x} = \frac{138}{106}$ | Oxygen to carbon ratio |
| $NO_3 : C$ | mol/mol | $\frac{4x+3y}{5x} = \frac{94.4}{106}$ | Nitrate to carbon ratio |
| $SO_4 : C$ | mol/mol | $\frac{106}{212}$ | sulphate to carbon ratio |
| **Secondary reaction parameters** | | | |
| $\gamma_{NH_4}$ | - | 0.9 | Fraction of $NH_4$ that is nitrified |
| $\gamma_{H_2S}$ | - | 0.95 | Fraction of $H_2S$ that is oxidised |
| $\gamma_{CH_4}$ | - | 0.99 | Fraction of $CH_4$ that is oxidised |
| **Adsorption coefficients** (Wang and Van Cappellen, 1996; Slomp et al., 1998) | | | |
| $K_{NH_4}$ | - | 1.4 | $NH_4$ adsorption coefficient |
| $K_{PO_4}^{ox}$, $\quad K_{PO_4}^{anox}$ | - | 200.0, 2.0 | $PO_4$ adsorption coefficient (oxic, anoxic) |
| **P related parameters** (Slomp et al., 1996) | | | |
| $k_s$ | yr$^{-1}$ | 94.9 | Rate constant for $PO_4$ sorption |
| $k_m$ | yr$^{-1}$ | 0.193 | Rate constant for Fe-bound P release |
| $k_a$ | yr$^{-1}$ | 0.365 | Rate constant for authigenic CFA precipitation |
| $PO_4{}^s$ | mol cm$^{-3}$ | $1 \cdot 10^{-9}$ | Equilibrium concentration. for P sorption |
| $FeP^\infty$ | mol cm$^{-3}$ | $1.99 \cdot 10^{-10}$ | Asymptotic concentration for Fe-bound P |
| $PO_4{}^a$ | mol cm$^{-3}$ | $3.7 \cdot 10^{-9}$ | Equilibrium conc. for authigenic P precipitation |





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
