# Peer review of "OMEN-SED(-RCM) (v1.1): A pseudo reactive continuum representation of organic matter degradation dynamics for OMEN-SED"

_Geoscientific Model Development, 2021_

## Author Comment (AC1)

Subject: Response to Reviewer's comments on BG-2021-4

5 Manuscript ID: gmd-2021-4 "OMEN-SED(-RCM) (v1.1): A pseudo reactive continuum representation of organic matter degradation dynamics for OMEN-SED"

Dear Reviewer 1,

10

On behalf of the co-authors, I want to thank you for the detailed and constructive review of our manuscript. In the following, we reply to each of the issues raised and explain how these will be addressed in the revised manuscript.

20 Sincerely yours,

Philip Pika and co-authors

**General comments**

- 25 The authors presented a multi-G approximation method, in which organic matter (OM) can be divided into a large number of classes with different degradation rate to represent a pseudo reactive continuum of OM, as an extension / further development of an existing analytical early diagenetic model (OMEN-SED) that is originally based on two reactive OM classes (2-G model). The approximation is based on two assumptions, namely (1) the
- 30 distribution of reactivity/degradation rate of OM in marine sediments can be reasonably described by a gamma function, and (2) the vertical OM distribution in sediments is in an equilibrium status (i.e. the temporal gradient of OM is zero at any depth in sediments) so that an analytical solution of the transport-reaction equation of OM can be derived. Because the proposed multi-G approach is based on the analytical solution, which does not require
- 35 solving the transport-reaction equation dynamically, therefore computational expense is not a hindering factor. This makes the proposed method appealing.

However, I have several major concerns:

**Comment 1:**

although the method seems to produce reasonable results, the authors did not provide
convincing arguments that the proposed method outperforms the original 2-G model;

**Response:**

We agree with the reviewer that the manuscript would benefit from a short discussion of the
 advantages and disadvantages of continuum and (multi-) 2-G models. We will include this in
 the introduction of the revised version.

However, we would like to emphasize that the aim of the new model development is not to "outperform" the original version, but rather to offer a choice between two, equally valid OM
degradation model formulations. When tuned to local conditions, both the original 2-G model and the RCM-approximation can perform equally well in reproducing comprehensive local porewater/sediment depth profiles and recycling fluxes, and both are very valuable approaches for estimating reaction rates in surface sediments (i.e. the top tens of centimeters). In addition, the 2G model approach also aligns well with the model description of the biological carbon pump (power law or (multi-)exponential equation) of most existing global biogeochemical models and earth system models.

However, for global scale applications, greater sediment depths or longer timescales, the continuum model has a number of advantages over the original 2G-Model:

- it captures the widely observed continuous decrease in OM reactivity with burial time/depth, while the 2G model converges to a constant apparent OM reactivity at depth (when the reactive OM-pool is consumed)
- 65 it merely requires constraining two free parameters, while, multi-G models, with n fractions, generally require the specification of 2n-1 parameters (the reactivity and the relative size for the n OM classes). Thus they are generally over-parameterized and difficult to constrain on a global scale.

70 We will clearly state these points in the revised introduction of the manuscript and also shortly discuss the advantages and disadvantages of each approach. Furthermore, we will make reference to relevant papers that discuss this topic more extensively (e.g., Manzoni and Porporato (2009); Forney and Rothman, 2012; Arndt et al., 2013; LaRowe et al., 2020).

**75 **Comment 2:**

the method is described in an unclear manner;

**Response:**

We will carefully revise the method section.

80 Please also see our responses to comments 3, 6, 7, 8 and 9 (Reviewer 1) as well as our response to comment 1 (Reviewer 2) for more details.

**Comment 3:**

85 while the method trys to fit observational vertical profiles in sediments, the boundary conditions needed for the model-data fit at some places do not reflect reality and bio-physcial laws; and

**Response:**

90 Here boundary conditions and model parameters are constrained as in Hülse et al. (2018). The only parameters that are tuned to fit the pore water profiles are the RCM parameters *a* and *v*. We will revise the text to clarify this.

Please also see our reply to comment 9 'specific comment on section 3.1' (Reviewer 1) for a more detailed explanation.

95

100

**Comment 4:**

the precondition for the validity of the approach, namely a zero temporal gradient of OM at any depth in sediments, can hardly be met in a dynamic environment. This makes the approach of limited use for coupling to dynamic models in which sedimentation of OM is variable, which is especially true for continental margins.

**Response:**

This is indeed one of the main limitations of OMEN-SED(-RCM)'s analytical model
approach (see section 5 in Hülse et al. 2018). OMEN-SED-(RCM) has been primarily
designed for the coupling to Earth System models and to investigate long term sediment
dynamics. In this context, the required assumption of steady-state is valid because the
variability in boundary conditions and fluxes is generally longer than the characteristic
timescales of the reaction-transport processes. However, the steady state assumption can be a

110 limitation for the model's applicability to shallow coastal environments.

Nevertheless, both, the previous and current version of OMEN-SED are able to reproduce observed porewater dynamics across different depositional environments ranging from the coastal to the deep ocean as evidenced by the model-data and model-model comparisons.

115 Especially the good agreement between OMEN-SED and the fully formulated numerical RTM (BRNS, Section 3.3) shows that the steady-state assumption is not a critical limitation of OMEN-SED. In addition, as outlined in section 5: "Scope of applicability and model

limitations" of Hülse et al. (2018) additional developments, such as adapting pseudo-transient dynamics will further facilitate the application of OMEN-SED to more dynamic environments.

**120 environments.**

We will re-iterate these points again in the discussion of the steady-state limitation in the revised manuscript.

**125 Specific comments:**

**Comment 5:**

\* The method (section 2) is not described clearly.

**130 **Response:**

We will thoroughly revise the model description section. Please also see our response to comments 6, 7, 8 and 9 as well as comment 1 (Reviewer 2).

**Comment 6:**

135 (a) From eq.6, it is stated that om(k,t) represents the probability density function that determines the amount of bulk OM with a reactivity between k and k+dk at time t. As om(k,t)is a probability density function, the sum of om(k,t) across all k at any speficic t should always be 1. However, this is not satisfied in eq.7, in which om(k,0) is dependent on OM(0). Please clarify this.

140

**Response:**

Correct, om(k,t)/OM(0) is actually the probability density function. We will clarify this in the revised version.

145

**Comment 7:**

(b) I am confused by the definition and use of k. k is supposed to indicate the reactivity of OM, which is a variable. So what is the justification of eq.8 that k is determined by a, v and sediment age? I understand that the latter three parameters at any specific depth are either

- 150 sediment age? I understand that the latter three parameters at any specific depth are either pre-described (e.g. v=0.15 in case studies) or derived by model-data fitting. This means that k is also fixed by these values, which is not variable any more. Further, how is the age of sediment layer at depth z derived? It seems that this quantity is another variable which needs to be solved in the method, in addition to a and v. This contradicts the statement and
- 155 conclusion that only a and v need to be solved. Please clarify.

**Response:**

In this case, k represents the apparent OM reactivity of the bulk OM mixture. It is indeed a function of a and v, as well as age(z) (or burial time) and thus varies with depth, z. The

160 age(z), in turn is a function of sedimentation rate, porosity (and bioturbation intensity). We agree that this part of the model description might be confusing and will therefore carefully revise these parts.

OMEN-SED(-RCM) does not solve the RCM (and thus Eq. (4) and/or age(z)) but applies a 500G approximation of the RCM which is informed by the initial distribution of OM compounds over the reactivity spectrum as calculated by the gamma-distribution.

The bulk OM reactivity,  $k_{total}(z)$ , of our RCM-approximation varies with depth and can be calculated as:

170
$$k_{total}(z) = \frac{\sum k_i * OM_i(z)}{OM_{total}(z)}$$

In the global application  $k_{total}(z)$  also varies spatially, because we apply a different set of *a*- $\nu$  values at each grid point. We will therefore omit eq. 4 and 8 as they are not relevant for our model approach and we will clarify this point.

175

**Comment 8:**

(c) Another parameterization of k using eq. 16 clearly violates the original relationship as mentioned in (b). Please justify the validity of the method if a different parameterization of k is used.

180

185

**Response:**

We recognize that this part of our manuscript might be confusing and thank the reviewer for pointing this out.

This part of the manuscript describes the parametrization of Thullner et al. (2009) and Hülse et al. (2018). They apply equation 16 to derive the first-order OM degradation rate constant

for their 1G model approach. In contrast, we here use a previously published a=f(w) relationship (Equation 15) to calculate

In contrast, we here use a previously published a=f(w) relationship (Equation 15) to calculate the sedimentation rate dependent apparent initial age of the OM mixture (parameter *a*). From the resulting initial OM distribution, we derive the discrete OM fractions and their respective first order OM degradation rate constants for the 500C approximation. The bulk OM

190 first-order OM degradation rate constants for the 500G approximation. The bulk OM reactivity at the SWI of our 500G approximation (k\_total in Table 2) is then calculated as:

$$k_{total}(z) = \frac{\sum k_i * OM_i(z)}{OM_{total}(z)}$$

195 We realize that reporting this equation in such a prominent manner leads to confusion and will thus revise this section accordingly.

**Comment 9:**

- 200 \* In the case study 3.1, although the free variables a, v are tuned that the model produces results close to observed sediment profiles, their setting has no mechanistic connection with other environmental variables, e.g. in Table 1, it is not clear why zbio is set to 0.01 cm at depth 585m, which means that there is no bioturbation at all, but then why Dbio has a non-zero value and how these parameters are related to the setting of a and v? Also it is not clear
- 205 why a has a very small value (corresponding to very small lifetime of OM, therefore quite labile component) for depth 2213 m. Compared to a very confusing setting in this case study, the setting in 3.2 (Table 2) seems more reasonable and respects reality.

**Response:**

210 For this set of simulations, all boundary conditions and parameter values are taken from Hülse et al. (2018), who adapted the boundary conditions and parameter values from the original publications where available or chose the default parameter values set in OMEN-SED, which in turn were constrained based on published values.

Hence the bioturbation coefficients for the Iberian margin sites (108, 2213 and 4298m) are
taken from Epping et al. (2002). The Santa Barbara site (585m) is characterized by anoxic
bottom waters (compare Reimers et al., 1990 and table 1). Therefore we set zbio to 0.01 cm
(i.e. no bioturbation). Db is set to a non-zero value for mathematical reasons. In some
equations underlying OMEN-SED, Db is a term in the denominator. As a consequence, we
set Db to a small non-zero value to avoid dividing by zero. But, the table actually contains a

typo. For the site at 585 m depth, Db is set to 1e-20 and not 0.02. We will correct this typo.

RCM parameters are constrained based on best-fit solutions to the porewater observations and are independent from the other boundary conditions listed in Table 1. The best-fit solution for the observed porewater and depth profiles at the deep site on the Iberian margin yields indeed a comparably low a value indicating an initial presence of comparably reactive

- 225 components. This is in line with the comparably high (k=0.1 yrs-1) first-order reaction rate constant of the more reactive pool determined by Hulse et al., 2018 and reflects the heterogeneity of OM quality on the spatial scale. Apparent OM reactivity is controlled by a complex interplay of environmental factors and OM deposited at 2000 meter depths (or more) and might be as reactive or even more reactive than OM deposited in the shallow
- 230 ocean (see e.g. Fig. 15a in Arndt et al., 2013). The Iberian margin is a highly productive and dynamic depositional environment that is characterized by the development of pronounced nepheloid layers in the area Nazaré Canyon (Epping et al., 2002). These nepheloid layers extend to considerably greater water depths and efficiently transport more reactive OM laterally down the slope. Thus the intense lateral transport of OM might explain the accurrence of comparably reactive OM at greater depths.
- 235 occurrence of comparably reactive OM at greater depths.

We will carefully revise the site-specific information.

**Comment 10:**

240 \* There is hardly justification for the validity of the approach in global application in section

3.3, as shown in Fig.6. In particular, the part that simulated OPD exceeds  $10^3$  mm is not supported by any observation.

**Response:**

- 245 In section 3.3, we aim to illustrate the model's ability to simulate diagenetic dynamics on the global scale. However, as pointed out in the manuscript itself, application of diagenetic models in data-poor areas, such as the global-scale, are currently limited by the lack of a general framework that would allow constraining OM degradation model parameters (e.g. see also Arndt et al., 2013).
- Here, we here use the weak relationship between parameter a and sedimentation rate that emerges from a compilation of previously published a values (see Arndt et al. 2013 for detailed information). Although this relationship captures the very broad global patterns in OM reactivity, it does by no means capture the full variability of OM reactivity across different depositional environments and thus diagenetic dynamics (including oxygen dynamics and OPD).

We agree that our results show that OMEN-SED(-RCM) tends to overpredict observed OPD. This can be partly explained with:

1) the limitations inherent to the OM degradation model parametrization on the global scale

The weak relationship between parameter a and sedimentation rate (w) that is applied to constrain parameter a for these simulations also tends to overpredict a (and thus underpredict OM reactivity) in shallower areas and thus also contributes to the mismatch between observations and simulation results.

265

260

 a bias in the observational data set towards shallower OPDs. In fact, deep OPD of several meters to kilometers have been widely observed in the central parts of the oceans (D'Hondt et al., 2015; Murray, J. W. & Grundmanis, 1980, Roy et al., 2012). Based on observations from the South Pacific gyres, D'Hondt et al., 270 2015 derive an empirical relationship between O2 penetration depth and sedimentation rate as well as sediment thickness to show that oxygen may be present throughout the entire sediment sequence in 15–44% of the Pacific and 9–37% of the global sea floor. These estimates are further supported by the absence of a sulfate-methane transition zone in these areas (Eggert et al. 2018 and see Fig. 3 LaRowe et al., 2020 for global maps: https://www.sciencedirect.com/science/article/abs/pii/S0012825219305720).

We will further discuss this mismatch in the revised manuscript and will also add simulation results from an OMEN-SED-RCM run with a re-scaled version of the *a*-w relationship that accounts for the overprediction of a in shallower areas.

Arndt, S., Jørgensen, B. B., LaRowe, D. E., Middelburg, J. J., Pancost, R. D., and Regnier, P. (2013). Quantifying the degradation of organic matter in marine sediments: A review and synthesis. Earth-Science Reviews, 123:53–86.

Arndt, S., and Regnier, P. A model for the benthic-pelagic coupling of silica in estuarine ecosystems: sensitivity analysis and system scale simulation. Biogeosciences 4, no. 3 (2007): 331-352.

Egger, M., Riedinger, N., Mogollón, J. M., and Jørgensen, B. B. (2018). Global diffusive fluxes of methane in marine sediments, Nature Geoscience, 11, 421–425.

Epping, E., Van Der Zee, C., Soetaert, K., and Helder, W. (2002). On the oxidation and burial of organic carbon in sediments of the Iberian Margin and Nazaré Canyon (NE Atlantic). Progress in Oceanography, 52(2-4):399–431.

295

Freitas, Felipe S., Philip A. Pika, Sabine Kasten, Bo B. Jørgensen, Jens Rassmann, Christophe Rabouille, Shaun Thomas, Henrik Sass, Richard D. Pancost, and Sandra Arndt. (2021). Advancing on large-scale trends of apparent organic matter reactivity in marine sediments and patterns of benthic carbon transformation. *Biogeosciences Discussions*: 1-64.

300

Glud, R. N. (2008). Oxygen dynamics of marine sediments. Marine Biology Research, 4(4):243–289.

D'Hondt, S., Inagaki, F., Zarikian, C. A., Abrams, L. J., Dubois, N., Engelhardt, T., Evans,

H., Ferdelman, T., Gribsholt, B., Harris, R. N., Hoppie, B. W., Hyun, J.-H., Kallmeyer, J., Kim, J., Lynch, J. E., McKinley, C. C., Mitsunobu, S., Morono, Y., Murray, R. W., Pockalny, R., Sauvage, J., Shimono, T., Shiraishi, F., Smith, D. C., Smith-Duque, C. E., Spivack, A. J., Steinsbu, B. O., Suzuki, Y., Szpak, M., Toffin, L., Uramoto, G., Yamaguchi, Y. T., Zhang, G.-l., Zhang, X.-H., and Ziebis, W. (2015). Presence of oxygen and aerobic communities
from sea floor to basement in deep-sea sediments. Nature Geoscience, 8(4):299–304.

Hülse, D., Arndt, S., Daines, S., Regnier, P., and Ridgwell, A. (2018). OMEN-SED 1.0: a novel, numerically efficient organic matter sediment diagenesis module for coupling to Earth system models. Geoscientific Model Development, 11(7):2649–2689.

315

LaRowe, D. E., Arndt, S., Bradley, J. A., Estes, E. R., Hoarfrost, A., Lang, S. Q., Lloyd, K. G., Mahmoudi, N., Orsi, W. D., Shah Walter, S. R., Steen, A. D., and Zhao, R. (2020). The fate of organic carbon in marine sediments - New insights from recent data and analysis. Earth-Science Reviews, 204(August 2019):103146.

320

Manzoni, Stefano, and Amilcare Porporato. (2009). Soil carbon and nitrogen mineralization: theory and models across scales. Soil Biology and Biochemistry 41, no. 7: 1355-1379.

Murray, J. W., & Grundmanis, V. (1980). Oxygen consumption in pelagic marine sediments. Science, 209(4464), 1527-1530.

Reimers, C. E., Lange, C. B., Tabak, M., & Bernhard, J. M. (1990). Seasonal spillover and varve formation in the Santa Barbara Basin, California. Limnology and Oceanography, 35(7), 1577-1585.

Roy, H., Kallmeyer, J., Adhikari, R. R., Pockalny, R., Jorgensen, B. B., and D'Hondt, S. (2012). Aerobic Microbial Respiration

Ruardij, P., and Van Raaphorst, W.. "Benthic nutrient regeneration in the ERSEM ecosystem model of the North Sea." Netherlands Journal of Sea Research 33.3-4 (1995): 453-483.

Thullner, M., Dale, A. W., and Regnier, P. (2009). Global-scale quantification of mineralization pathways in marine sediments: A reaction-transport modeling approach. Geochemistry, Geophysics, Geosystems, 10(10).

---

## Author Comment (AC2)

Date: July 8$^{th}$, 2021

Subject: Response to Reviewer's comments on BG-2021-4

Manuscript ID: gmd-2021-4 "OMEN-SED(-RCM) (v1.1): A pseudo reactive continuum representation of organic matter degradation dynamics for OMEN-SED"

Dear Bernard Boudreau,

On behalf of the co-authors, I want to thank you for the detailed and constructive review of our manuscript. In the following, we reply to each of the issues raised and explain how these will be
addressed in the revised manuscript.

Sincerely yours,

Philip Pika and co-authors

*This paper reports a means by which the OMEN-SED diagenesis program (Hülse et al., 2018,*
*GMD 11), which uses 2-G organic matter (OM) decay kinetics, can be altered to approximate*
*a reactive continuum decay model, instead. The interest in this modification lies in better*
*approximating the observed decrease in OM reactivity with depth in sediments, which is a*
*desirable goal. I do have four comments on the execution of this paper, one of which reflects*
*one that I made on a recent related submission by some of these authors, i.e., Freitas et al.*
*(2020, BG-2020-435).*

**Comment:**

*1) Equations (4) and (8) in the Introduction and on page 7 are strictly valid only in non-*
*bioturbated sediments, where there is a simple relationship between time and depth, a point*
*that is only briefly mentioned in the present manuscript. These equations are stated and*
*applied without any serious caveat. The correct expression with bioturbation is given in*
*Boudreau and Ruddick (1991), their Equation (49), and it is not in an easily implemented*
*form, as needed here.*
*No one has adequately explored the effects of applying Equation (8) to a bioturbated*
*sediment. In Table 1 of Boudreau and Ruddick (1991), there are 6 applications of Equation*
*(8) to non-bioturbated or negligibly bioturbated sediments. These generate v values between*
*0.05 and 0.3; the Peru Margin sediments in that same table are almost certainly bioturbated*
*and have associated v values closer to unity (0.8 and 0.91). This dichotomy of results may*
*indicate the effects of improperly dealing with bioturbation, but I really cannot affirm this. I*
*also recognize that the actual calculations are made with a discrete numerical*
*approximation, which removes much of the mathematical challenge, but also assumes a v*
*characteristic of a non-mixed sediment; this may create unintentional modifications to the*
*calculated "a" value.*
*Further to this point, om(k,0) in Equation (7) is said to be the initial distribution of*
*reactivities. In a non-mixed sediment, that distribution is exclusively altered by the decay of*
*components with time. In a bioturbated sediment, "older" and "newer" organic matter*
*components are mixed together, so that in leaving the mixed layer, the organic matter*
*reactivity distribution is different than om(k,0) at the SWI. Thus, the mathematics get even*
*more demanding.*
*Progress in global modelling is needed to meet the challenges of global change; I appreciate*
*that; corners will be cut by necessity. However, explicitly recognizing and stating those*
*shortcuts is not only desirable, bit is essential.*

**Response:**
We really appreciate the points raised and  agree that the method section needs further
clarification and explanation. More specifically we will make the following points clearer:

Although, equations (4) and (8) are strictly valid only in non- bioturbated sediments, they, as
well as other, empirical continuum models (Middelburg et al., 1989, Jørgensen et al., 2010)
that use a similar expression to describe the depth evolution of apparent OM reactivity have
been successfully applied across a wide range of depositional environments (see
compilation of published studies in Arndt et al., 2013). However, like many other factors
that are not explicitly accounted for (e.g. microbial community structure, temperature,
thermodynamic inhibition, TEA availability, …) the effects of bioturbation (not only transport, but also priming, stabilization of OM, ... ) will be implicitly accounted for in the parameter values, thus explaining the reported variability in model parameters.

In addition, as pointed out by the reviewer, we here use a discrete approximation of the continuum model that is merely constrained on the initial distribution om(k,0), but makes no assumptions about the evolution of bulk OM reactivity in the bioturbated zone or how *a-v* characteristics would relate to it.

Within this discrete approximation (i.e. the 500 G model), the evolution of the apparent bulk OM reactivity within the bioturbated zone (and below) is  not only determined by the specific consumption rates of the 500 OM pools (G), but also by their transport via bioturbation and advection. The 500 G model approximation thus accounts for the mixing of "older" and "newer" organic matter components (as long as the assumed steady state conditions apply). In leaving the mixed layer, the organic matter reactivity distribution is thus indeed different than om(k,0) at the SWI and would be equivalent to om(k, t_apparent) with t_apparent being the apparent age of the OM that is determined by transport and degradation processes.

We will include a section that discusses the caveats, simplifications and limitations of our approach in more detail. Here we will also note that, in reality, the distribution of OM reactivities are likely to be more complex and that our choice of the specific mathematical approximation represents a necessary simplification in the first place (as has been noted by the reviewer before, e.g. in Boudreau & Ruddick, 1991). We will also revise the equations stated in our Introduction and Model Description (especially equations 4, 7 and 8) and adapt them so they reflect our specific discrete approximation of the continuum model better. In particular we will omit Equations (4) and (8) from the manuscript as they are not relevant for our approach.

**Comment:**

*2) The methodology of this paper and of Hülse et al. (2018) is to create a diagenetic module that can be solved analytically and to calculate the resulting integration constants by fast linear matrix methods (see section 2.3.1 of Hülse et al., 2018). I am confused about this second part, as there is insufficient detail in either paper to see how the latter can be accomplished. Take sulfate in its reduction zone. The analytical solution will be a sum of*

*exponentials, two of which will be multiplied by unknown integration constants. Boundary conditions at the top of the zone and at the base of the zone will allow calculation of the two constants, and these resulting equations will indeed be linear in the two integration constants. However, the position of the base of the sulfate reduction zone is a further unknown, a point that was extensively discussed in Boudreau and Westrich (1984, GCA 48,*

*2503-2516). A third boundary condition is needed to determine the unknown zSO4.*

*Hülse et al. (2018) do supply a third equation, and so it should be in the present paper too. What puzzles me is that this third equation, as in Boudreau and Westrich (1984), is a nonlinear function of zSO4. Thus, linear matrix methods cannot be employed to arrive at a*

*solution. What have the authors done to avoid this problem, which seems fundamental to their paper?*

**Response:**

The explanation for how to solve the dynamically changing penetration depth problem is explained in Hülse et al. (2018). However, it is actually included in the subsections for the specific terminal electron rather than in section 2.3.1.Briefly, taking the example of the reviewer, i.e. sulfate in its reduction zone (compare 2.2.4 of Hülse et al., 2018):

OMEN-SED iteratively solves for zSO4 by first testing if SO4 is still available at zmax (i.e. SO4

(zmax) > 0). In that case, zSO4 is not an unknown anymore (as zSO4 = zmax) and just the zero diffusive flux boundary condition at the lower boundary is sufficient to solve the diagenetic equation (compare (8.1) in Table 5 of Hülse et al., 2018). In the case zSO4 < zmax (i.e. zSO4 is unknown) an extra boundary condition is needed. Here, OMEN-SED requires (1) zero SO4 concentration at zSO4 and (2) the SO4 diffusive flux at zSO4 must equal the flux of methane from below that is reoxidized (compare 8.2 in Table 5 of Hülse et al., 2018). OMEN-SED then iteratively solves for zSO4.

We will include a specific reference (including page number) to the relevant section of the original OMEN-SED publication in a relevant section of the revised model description.

**Comment:**

*3) I fail to agree that the model predictions and observations provided in Figure 6 "generally agree well with observations and capture also the widely observed increase in OPD with water depths (see Fig. 6)." The model in this paper generally overpredicts the OPD, more often by far more than an order of magnitude. The data appear to have an upper limit of 1000 mm with ocean depth, which is not present in the model predictions. The obvious discrepancies are not discussed in the paper. They really need to be. Is the "observed" 1000 mm limit an artifact or real? Why the common overprediction of the OPD? There is something valuable to be learned from negative results!*

**Response:**

Also see response to Reviewer 1

We agree that the presented model generally overpredicts the observed OPD. We will discuss and address this shortcoming in the revised manuscript. The mismatch between model results and observations can be partly explained by

   1) A bias in the observational data set towards shallower OPD

The upper limit of 1000mm in the observed OPD is likely an artefact of the data selected here, since oxygen is known to penetrate several meters (Glud, 2008), or even down to 60m (see D'Hondt et al., 2015; Roy et al., 2012), in the oligotrophic gyres of the South Pacific. The observational data set does not include these deep OPD of several meters to kilometers that have been observed in the central parts of the oceans (D'Hondt et al., 2015; Murray, J. W. & Grundmanis, 1980, Roy et al., 2012). We will update our database and/or discuss the bias in the revised text.

   2) two simplifications/limitations in the model configuration and boundary conditions

First, the applied OM concentration at the sediment-water interface (SWI) (Seiter et al. (2004) is actually more representative for the mean OM concentration in the upper 5cm of the sediment. It thus represents a lower limit for the actual SWI concentration. Second, the applied a-w parameterization used to calculate the average life-time of the more reactive OM compounds represents only a weak relationship ($R^2$ = 0.46) as it is only based on a limited number of observations (see Fig. 15b of Arndt et al., 2013). The resulting a-values following the parameterization are in the range of 34 – 2239 yrs. with a mean value of 1240 yrs. ($\sigma$ = 955 yrs.) and thus likely more representative for an OM mixture of lower reactivity. Both limitations result in lower OM degradation rates and thus an overestimation of the OPD.

We will include additional results in the revised manuscript using a rescaled a-w relationship that yields lower *a*-values, while still following the widely observed inverse trend between *a* and sediment accumulation rate (*w*) (e.g. Middelburg, 1989; Boudreau and Ruddick, 1991). The new results should agree better with the observations. We agree with the reviewer that there is something valuable to learn from the negative results, namely that the a-w parameterization generally results in rather low OM reactivities. This will be discussed in the revised manuscript alongside other limitations of this specific modeling exercise (i.e. the global maps of SWI TOC concentrations and sediment accumulation rates certainly have flaws, the distribution of OM compounds is certainly more complex in reality and site specific peculiarities are not represented in our approach). Thus some degree of mismatch is to be expected (while the reviewer is correct that our previous results generally overpredict the OPD).

The upper limit of 1000m in the observed OPD is likely an artefact of the data selected here, since oxygen is known to penetrate several meters (Glud, 2008), or even down to 60m (see D'Hondt et al., 2015; Murray & Grundmanis, 1980; Roy et al., 2012), in the oligotrophic gyres of the South Pacific. We will try to update our database and/or note the artefact in the revised text.

**Comment:**

*4) Finally, as much as Equation (15) might prove valuable, it is repeated as Equation (17) with a discussion that is similar. Is this duplication needed or am I missing something?*

**Response:**

Agreed with being a duplication and unnecessary, and will be removed and Equation (15) will then be referenced in the Global application (3.3).

Arndt, S., Jørgensen, B. B., LaRowe, D. E., Middelburg, J. J., Pancost, R. D., and Regnier, P.
(2013). Quantifying the degradation of organic matter in marine sediments: A review and
synthesis. Earth-Science Reviews, 123:53–86.

D'Hondt, S., Inagaki, F., Zarikian, C. A., Abrams, L. J., Dubois, N., Engelhardt, T., Evans, H.,
Ferdelman, T., Gribsholt, B., Harris, R. N., Hoppie, B. W., Hyun, J.-H., Kallmeyer, J., Kim, J.,
Lynch, J. E., McKinley, C. C., Mitsunobu, S., Morono, Y., Murray, R. W., Pockalny, R., Sauvage,
J., Shimono, T., Shiraishi, F., Smith, D. C., Smith-Duque, C. E., Spivack, A. J., Steinsbu, B. O.,
Suzuki, Y., Szpak, M., Toffin, L., Uramoto, G., Yamaguchi, Y. T., Zhang, G.-l., Zhang, X.-H., and
Ziebis, W. (2015). Presence of oxygen and aerobic communities from sea floor to basement
in deep-sea sediments. Nature Geoscience, 8(4):299–304.

Murray, J. W., & Grundmanis, V. (1980). Oxygen consumption in pelagic marine sediments.
Science, 209(4464), 1527-1530.

Glud, R. N. (2008). Oxygen dynamics of marine sediments. Marine Biology Research,
4(4):243– 289.

Roy, H., Kallmeyer, J., Adhikari, R. R., Pockalny, R., Jorgensen, B. B., and D'Hondt, S. (2012).
Aerobic Microbial Respiration in 86-Million-Year-Old Deep-Sea Red Clay. Science,
336(6083):922– 925.

---

## Author Response (AR1)

**OMEN-SED(-RCM) (v1.1):**
**A pseudo reactive continuum representation of organic matter degradation dynamics for OMEN-SED**

**Author's Response**

Philip Pika, Dominik Hülse, Sandra Arndt
30th August 2021

Dear Editor,
please find below our point to point listing of the changes made to the manuscript in response to the referees' comments and suggestions. A *latexdiff* version of the manuscript highlighting the insertions and deletions in the text has also been uploaded alongside the revised manuscript.
We hope the manuscript is now acceptable for publication.

Best regards,
Philip Pika

**Revisions in response to comments by Anonymous Referee #1**

**General comments**

*The authors presented a multi-G approximation method, in which organic matter (OM) can be divided into a large number of classes with different degradation rate to represent a pseudo reactive continuum of OM, as an extension / further development of an existing analytical early diagenetic model (OMEN-SED) that is originally based on two reactive OM classes (2-G model). The approximation is based on two assumptions, namely (1) the distribution of reactivity/degradation rate of OM in marine sediments can be reasonably described by a gamma function, and (2) the vertical OM distribution in sediments is in an equilibrium status (i.e. the temporal gradient of OM is zero at any depth in sediments) so that an analytical solution of the transport-reaction equation of OM can be derived. Because the proposed multi-G approach is based on the analytical solution, which does not require solving the transport-reaction equation dynamically, therefore computational expense is not a hindering factor. This makes the proposed method appealing.*

*However, I have several major concerns:*

**Comment 1:**
*although the method seems to produce reasonable results, the authors did not provide convincing arguments that the proposed method outperforms the original 2-G model;*

**Response:**

We would like to re-iterate again that the aim of the new model development is not to "outperform" the original version, but rather to offer a choice between two, equally valid OM degradation model formulations. Each of these formulations has specific advantages and disadvantages and we have added short discussion of the advantages and disadvantages of continuum and (multi-) 2-G models in the introduction (p. 4, l. 96-111). Furthermore, in the revised introduction, we also reference relevant papers that discuss this topic more extensively

(e.g., Manzoni and Porporato (2009); Forney and Rothman, 2012; Arndt et al., 2013; LaRowe et al., 2020).

**Comment 2:**
*the method is described in an unclear manner;*

**Response:**
- We clarified the methods section by removing unnecessary equations (eq. 4 and 8), adding further explanation (p. 8 l. 186-191) and/or rewriting parts of the methods (p. 7-8 l. 175-191). Please also see our detailed responses to the more specific comments 3, 6, 7, 8 and 9 (Reviewer 1) as well as our response to comment 1 (Reviewer 2) for more information.

**Comment 3:**
*while the method trys to fit observational vertical profiles in sediments, the boundary conditions needed for the model-data fit at some places do not reflect reality and bio-physcial laws; and*

**Response:**
- Please note that boundary conditions and model parameters are constrained as in Hülse et al. (2018). The only parameters that are tuned to fit the pore water profiles are the RCM parameters a and v. In the revised manuscript, we clarified further which parameters we tune here and where we derive other constrains on boundary conditions and model parameters. Please also see our reply to comment 9 'specific comment on section 3.1' (Reviewer 1) for a more details.

**Comment 4:**
*the precondition for the validity of the approach, namely a zero temporal gradient of OM at any depth in sediments, can hardly be met in a dynamic environment. This makes the approach of limited use for coupling to dynamic models in which sedimentation of OM is variable, which is especially true for continental margins.*

**Response:**
- OMEN-SED-(RCM) has been primarily designed for the coupling to Earth System models and to investigate long term sediment dynamics. In this context, the required assumption of steady-state is valid because the variability in boundary conditions and fluxes is generally longer than the characteristic timescales of the reaction–transport processes. However, the steady state assumption can be a limitation for the model's applicability to shallow coastal environments. We added a paragraph addressing this limitation (p.6 l. 146-149) and also refer to the relevant section in Hülse et al., 2018, where additional model limitations are addressed.

**Specific comments:**

**Comment 5:**
*\* The method (section 2) is not described clearly.*

**Response:**
- We clarified the methods section by removing unnecessary equations (eq. 4 and 8), adding further explanation (p. 8 l. 185-191) and/or rewriting parts of the methods (p. 7-8 l. 175-191). Please also see our response to comments 6, 7, 8 and 9 as well as comment 1 (Reviewer 2).

**Comment 6:**
*(a) From eq.6, it is stated that om(k,t) represents the probability density function that determines the amount of bulk OM with a reactivity between k and k+dk at time t. As om (k,t) is a probability density function, the sum of om (k,t) across all k at any speficic t should always be 1. However, this is not satisfied in eq.7, in which om (k,0) is dependent on OM(0). Please clarify this.*

**Response:**
- We removed the term "probability density function", as it is indeed confusing.

**Comment 7:**
*(b) I am confused by the definition and use of k. k is supposed to indicate the reactivity of OM, which is a variable. So what is the justification of eq.8 that k is determined by a, v and sediment age? I understand that the latter three parameters at any specific depth are either pre-described (e.g. v=0.15 in case studies) or derived by model-data fitting. This means that k is also fixed by these values, which is not variable any more. Further, how is the age of sediment layer at depth z derived? It seems that this quantity is another variable which needs to be solved in the method, in addition to a and v. This contradicts the statement and conclusion that only a and v need to be solved. Please clarify.*

**Response:**
In this case, k represents the apparent OM reactivity of the bulk OM mixture. It is indeed a function of a and $\nu$, as well as age(z) (or burial time) and thus varies with depth, z. The age(z), in turn is a function of sedimentation rate, porosity (and bioturbation intensity). We carefully revised these parts. More specifically:

- We omitted eq. 4 and 8 as they are not relevant for our model approach
- We added a new equation for the bulk OM reactivity k(z) (i.e. Eq. 6), and a better explanation of our RCM approximation (p. 8 l. 175-191).

**Comment 8:**
*(c) Another parameterization of k using eq. 16 clearly violates the original relationship as mentioned in (b). Please justify the validity of the method if a different parameterization of k is used.*

**Response:**

This part of the manuscript describes the parametrization of Thullner et al. (2009) and Hülse et al. (2018). They apply equation 16 to derive the first-order OM degradation rate constant for their 1G model approach. We, in contrast, use a previously published a=f(w) relationship (Equation 15) to calculate the sedimentation rate dependent apparent initial age of the OM mixture (parameter a). We recognize that this part of our manuscript might be confusing and, therefore, removed Eq. 16.

**Comment 9:**
*\* In the case study 3.1, although the free variables a, v are tuned that the model produces results close to observed sediment profiles, their setting has no mechanistic connection with other environmental variables, e.g. in Table 1, it is not clear why zbio is set to 0.01 cm at depth 585m, which means that there is no bioturbation at all, but then why Dbio has a non-zero value and how these parameters are related to the setting of a and v? Also it is not clear why a has a very small value (corresponding to very small lifetime of OM, therefore quite labile component) for depth 2213 m. Compared to a very confusing setting in this case study, the setting in 3.2 (Table 2) seems more reasonable and respects reality.*

**Response:**
- Corrected typo in table 1. For the site at 585 m depth, Db is set to 1e-20 and not 0.02.

**Comment 10:**
*\* There is hardly justification for the validity of the approach in global application in section 3.3, as shown in Fig.6. In particular, the part that simulated OPD exceeds $10^3$ mm is not supported by any observation.*

**Response:**
In section 3.3, we aim to illustrate the model's ability to simulate diagenetic dynamics on the global scale. However, as pointed out in the manuscript itself, application of diagenetic models in data-poor areas, such as the global-scale, are currently limited by the lack of a general framework that would allow constraining OM degradation model parameters (e.g. see also Arndt et al., 2013). In addition, biases in observational data sets such as the bias towards settings with shallow OPD pointed out in our reply further affect global simulations.

- We now discus the mismatch between observed and simulated OPD and DOU values
- We added new simulation results from an OMEN-SED-RCM run with a re-scaled version of the *a*-w relationship that accounts for the overprediction of a in shallower areas, see figures 5 b, d, f) and 6 b)
- Updated section's methods accordingly (p. 20 l. 375-382)
- Expanded results/discussion according to the new results (p. 20 l. 384-419)

**Revisions in response to comments by Bernard Boudreau**

*This paper reports a means by which the OMEN-SED diagenesis program (Hülse et al., 2018, GMD 11), which uses 2-G organic matter (OM) decay kinetics, can be altered to approximate a reactive continuum decay model, instead. The interest in this modification lies in better approximating the observed decrease in OM reactivity with depth in sediments, which is a desirable goal. I do have four comments on the execution of this paper, one of which reflects one that I made on a recent related submission by some of these authors, i.e., Freitas et al. (2020, BG-2020-435).*

**Comment:**

*1) Equations (4) and (8) in the Introduction and on page 7 are strictly valid only in non-bioturbated sediments, where there is a simple relationship between time and depth, a point that is only briefly mentioned in the present manuscript. These equations are stated and applied without any serious caveat. The correct expression with bioturbation is given in Boudreau and Ruddick (1991), their Equation (49), and it is not in an easily implemented form, as needed here.*

*No one has adequately explored the effects of applying Equation (8) to a bioturbated sediment. In Table 1 of Boudreau and Ruddick (1991), there are 6 applications of Equation (8) to non-bioturbated or negligibly bioturbated sediments. These generate v values between 0.05 and 0.3; the Peru Margin sediments in that same table are almost certainly bioturbated and have associated v values closer to unity (0.8 and 0.91). This dichotomy of results may indicate the effects of improperly dealing with bioturbation, but I really cannot affirm this. I also recognize that the actual calculations are made with a discrete numerical approximation, which removes much of the mathematical challenge, but also assumes a v characteristic of a non-mixed sediment; this may create unintentional modifications to the calculated "a" value.*

*Further to this point, om(k,0) in Equation (7) is said to be the initial distribution of reactivities. In a non-mixed sediment, that distribution is exclusively altered by the decay of components with time. In a bioturbated sediment, "older" and "newer" organic matter components are mixed together, so that in leaving the mixed layer, the organic matter reactivity distribution is different than om(k,0) at the SWI. Thus, the mathematics get even more demanding.*

*Progress in global modelling is needed to meet the challenges of global change; I appreciate that; corners will be cut by necessity. However, explicitly recognizing and stating those shortcuts is not only desirable, bit is essential.*

**Response:**
We carefully revised the following aspects of the methods section:
  - Revised the equations stated in our Introduction and Model Description (removed Eq. 4 and 8)
  - Adapted text in Methods to clarify the discrete approximation of the continuum model (p.8 l. 175-191)

**Comment:**

*2) The methodology of this paper and of Hülse et al. (2018) is to create a diagenetic module that can be solved analytically and to calculate the resulting integration constants by fast linear matrix methods (see section 2.3.1 of Hülse et al., 2018). I am confused about this second part, as there is insufficient detail in either paper to see how the latter can be accomplished. Take sulfate in its reduction zone. The analytical solution will be a sum of exponentials, two of which will be multiplied by unknown integration constants. Boundary conditions at the top of the zone and at the base of the zone will allow calculation of the two constants, and these resulting equations will indeed be linear in the two integration constants. However, the position of the base of the sulfate reduction zone is a further unknown, a point that was extensively discussed in Boudreau and Westrich (1984, GCA 48, 2503-2516). A third boundary condition is needed to determine the unknown zSO4.*

*Hülse et al. (2018) do supply a third equation, and so it should be in the present paper too. What puzzles me is that this third equation, as in Boudreau and Westrich (1984), is a nonlinear function of zSO4. Thus, linear matrix methods cannot be employed to arrive at a solution. What have the authors done to avoid this problem, which seems fundamental to their paper?*

**Response:**

Hülse et al. (2018) provides an explanation for how to solve the dynamically changing penetration depth problem in the subsections for the specific terminal electron rather than in section 2.3.1. We included a specific reference to the relevant section of the original OMEN-SED publication in the revised manuscript (p. 6, l. 145-149).

**Comment:**

*3) I fail to agree that the model predictions and observations provided in Figure 6 "generally agree well with observations and capture also the widely observed increase in OPD with water depths (see Fig. 6)." The model in this paper generally overpredicts the OPD, more often by far more than an order of magnitude. The data appear to have an upper limit of 1000 mm with ocean depth, which is not present in the model predictions. The obvious discrepancies are not discussed in the paper. They really need to be. Is the "observed" 1000 mm limit an artifact or real? Why the common overprediction of the OPD? There is something valuable to be learned from negative results!*

**Response:**

Also see response to Reviewer 1. To clarify these points we:
- Discussed mismatch between observed and modelled OPD and DOU values
- Added new simulation results from an OMEN-SED-RCM run with a re-scaled version of the *a*-w relationship that accounts for the overprediction of a in shallower areas, see figures 5 b, d, f) and 6 b)
- Updated section's methods accordingly
- Expanded results/discussion according to the new results

**Comment:**

*4) Finally, as much as Equation (15) might prove valuable, it is repeated as Equation (17) with a discussion that is similar. Is this duplication needed or am I missing something?*

**Response:**
- Removed Eq. 17